# Pseudo-Labeling Optimization Based Ensemble Semi-Supervised Soft Sensor in the Process Industry

**DOI:** 10.3390/s21248471

**Published:** 2021-12-19

**Authors:** Youwei Li, Huaiping Jin, Shoulong Dong, Biao Yang, Xiangguang Chen

**Affiliations:** 1Yunnan Key Laboratory of Computer Technologies Application, Kunming 650500, China; 20192104045@stu.kust.edu.cn (Y.L.); biaoykmust@kust.edu.cn (B.Y.); 2Department of Automation, Faculty of Information Engineering and Automation, Kunming University of Science and Technology, Kunming 650500, China; 3Department of Chemical Engineering, School of Chemistry and Chemical Engineering, Beijing Institute of Technology, Beijing 100081, China; sldong@bit.edu.cn (S.D.); xgc1@bit.edu.cn (X.C.)

**Keywords:** soft sensor, unlabeled data, label scarcity, semi-supervised learning, ensemble learning, pseudo labeling, evolutionary optimization, negative correlation learning, extreme learning machine

## Abstract

Nowadays, soft sensor techniques have become promising solutions for enabling real-time estimation of difficult-to-measure quality variables in industrial processes. However, labeled data are often scarce in many real-world applications, which poses a significant challenge when building accurate soft sensor models. Therefore, this paper proposes a novel semi-supervised soft sensor method, referred to as ensemble semi-supervised negative correlation learning extreme learning machine (EnSSNCLELM), for industrial processes with limited labeled data. First, an improved supervised regression algorithm called NCLELM is developed, by integrating the philosophy of negative correlation learning into extreme learning machine (ELM). Then, with NCLELM as the base learning technique, a multi-learner pseudo-labeling optimization approach is proposed, by converting the estimation of pseudo labels as an explicit optimization problem, in order to obtain high-confidence pseudo-labeled data. Furthermore, a set of diverse semi-supervised NCLELM models (SSNCLELM) are developed from different enlarged labeled sets, which are obtained by combining the labeled and pseudo-labeled training data. Finally, those SSNCLELM models whose prediction accuracies were not worse than their supervised counterparts were combined using a stacking strategy. The proposed method can not only exploit both labeled and unlabeled data, but also combine the merits of semi-supervised and ensemble learning paradigms, thereby providing superior predictions over traditional supervised and semi-supervised soft sensor methods. The effectiveness and superiority of the proposed method were demonstrated through two chemical applications.

## 1. Introduction

Modern industrial processes are equipped with a large number of measurement devices, in order to allow the implementation of advanced monitoring, optimization, and control of the production process. However, many crucial quality variables in industrial process are difficult to measure online, due to the lack of reliable hardware sensors or the high investment in the purchase and maintenance of apparatuses. To tackle this problem, soft sensor technology, as a promising indirect measurement tool, has been proposed, to enable real-time estimations of difficult-to-measure process variables [1,2]. The basis of a soft sensor is to build a mathematical model describing the relationship between the difficult-to-measure target variable and the easy-to-measure secondary variables, and then perform online estimation for the query data, based on the built predictive model. Generally, soft sensors can be divided into two categories: first principle, and data-driven methods. The former method type requires deep physical and chemical knowledge, which is often impossible in many real-world applications. Alternatively, data-driven methods, only relying on historical process information and data analytics techniques, have gained great popularity in soft sensor applications over the past two decades [3,4,5,6,7,8].

In the context of data-driven soft sensor modeling, having sufficient process data is of great importance to guarantee the generalization capability of the model. Typically, a sample for model development can be split into two parts, i.e., attributes (also termed as inputs) and label (also termed as output), which correspond to the easily obtained process variables and the target variable to be estimated, respectively. Thus, the modeling samples with labels are called labeled data, while those lacking labels are called unlabeled data. However, the insufficiency of data information is very common in practical applications, which makes it difficult to establish an accurate data-driven model. Roughly speaking, there are two common data deficiency issues: (i) both input and output information are insufficient, and (ii) unlabeled data are sufficient, but labeled data are scarce. One popular approach to deal with the first scenario is virtual sample generation (VSG) [9,10] methods, which are applied to create virtual samples, using the original small data set. Another solution to this problem is transfer learning [11,12,13], which enables improving the model performance by sharing the relevant information from other domains with rich data. As for the second scenario, active learning [14,15] has been employed to provide the labels for the most informative unlabeled data, using an oracle such as a human annotator; thus, enlarging the labeled modeling data. Alternatively, semi-supervised learning (SSL) [16,17,18,19,20] has been widely used to enhance model prediction performance, by leveraging both labeled and unlabeled data. In practice, there are usually abundant unlabeled data, but limited labeled data in industrial processes, which has been the main bottleneck of building accurate soft sensor models. Thus, this paper focuses on semi-supervised soft sensor development.

As an important branch of machine learning, SSL, has undergone a long history and has become a hot topic [16,17,18,19,20]. SSL algorithms can effectively exploit information from both labeled and unlabeled data to build models whose accuracies are usually better than those obtained from only using the limited labeled data. So far, five typical classes of SSL algorithms have emerged [16]: generative models, semi-supervised support vector machines (S3VM), graph-based methods, self-training, and co-training. However, many of the current SSL methods mainly focus on the prediction of discrete variables in classification problems, such as imaging analysis and processing, natural language processing, and speech recognition [17]. Unfortunately, it is difficult or impossible to transfer these methods to the regression problems directly, as is often the case in soft sensor applications, which mainly concerns predictions of continuous target variables. Hence, research on semi-supervised soft sensors has gained considerable attention in recent years. A detailed review and discussions on this topic can be found in reference [18]. Here, we only outline some representative research:(1)Probabilistic generative models: This type of method aims to improve the estimation accuracy of the underlying distribution of the input space by introducing abundant unlabeled data. These methods assume that all samples, labeled and unlabeled, are generated from the same underlying model. The main difference among the different generative methods lies in the underlying assumptions. For example, Ge and Song [21] proposed a semi-supervised principal component regression (PCR) model, using a probabilistic method. The method first formulates a generative model structure according to the traditional PCR model. Then, by assuming that both probability density functions of the principal component and the process noise are Gaussian, the optimal parameters with respect to the data distribution are optimized by maximizing a likelihood function using an expectation-maximum algorithm, where supervised and unsupervised objectives are involved. Similarly, a semi-supervised probabilistic partial least squares regression model was developed for soft sensor modeling [22]. However, the assumption of Gaussian distribution is often ill-conditioned for industrial processes with multiple modes, where the process data usually exhibit non-Gaussian characteristics. To tackle this issue, non-Gaussian distribution based assumptions have been introduced to build generative soft sensor models, such as a mixture semi-supervised probabilistic PCR model [23], semi-supervised Gaussian mixture regression [24], semi-supervised Dirichlet process mixture of Gaussians [25], semi-supervised mixture of latent factor analysis models [26], and Student’s-t mixture regression [27]. Overall, the key to building an accurate generative model lies in accurate model assumptions, which are often difficult to determine without sufficient reliable domain knowledge.(2)Graph based methods: Methods of this type are based on manifold assumptions and require constructing a semi-labeled graph to assure the label smoothness over the graph. To this end, one needs to define a graph where the nodes denote labeled and unlabeled samples, and where the edges connect two nodes if their corresponding samples are highly similar. One typical example of such method is the label propagation method [28], originally proposed for addressing classification problems. As for graph based SSL soft sensors, it is a common practice to embed the graph based regularization into the cost function based on traditional supervised regression techniques. For example, two semi-supervised soft sensors were developed, by integrating the extreme learning machine (ELM) method and the graph Laplacian regularization into a unified modeling framework for industrial Mooney viscosity prediction [29,30]. Similarly, Yan et al. [31] developed a semi-supervised Gaussian process regression (GPR) for quality prediction, by using a semi-supervised covariance function, which was defined by introducing the manifold information into the traditional covariance function. Moreover, to enhance the model performance for handling complex process characteristics, as well as exploiting unlabeled process data, Yao and Ge [32] proposed a semi-supervised deep learning model for soft sensor development. First, it implements unsupervised feature extraction through an autoencoder with a deep network structure. Then, ELM is utilized for regression, by introducing manifold regularization. In addition, Yan et al. [33] proposed a semi-supervised deep neural regression network with manifold embedding for soft sensor modeling.(3)Representation learning based methods: A general strategy for these methods is to use unlabeled data to assist in extracting abstract latent features of the input data. The most common techniques for this purpose are deep learning methods [34], such as convolutional neural networks, deep belief networks (DBN), long/short-term memory neural networks, and a large variety of autoencoders. Besides its strong representation ability, deep learning is inherently semi-supervised and, thus, can effectively exploit all available process data. As an early attempt, Shang et al. [35] employed a DBN to build soft sensors for estimating the heavy diesel 95% cut-point of a crude distillation unit (CDU). However, traditional representation techniques are mainly implemented in an unsupervised manner, where the output variable information is ignored. To address this issue, several research works have focused on exploring semi-supervised representation learning techniques. For instance, Yan et al. [36] proposed a deep relevant representation learning approach based on a stacked autoencoder, which conducts a mutual information analysis between the representations and the output variable in each layer. Similar research has also been reported in references [37,38,39].(4)Self-labeled methods [40]: The core of such methods is extending the labeled training set by adding high-confidence pseudo-labeled data. In such a modeling framework, one or more predictive models are first trained with labeled data only, and then refined using the extended labeled set through iterative learning. Two representative examples are self-training [41], and co-training [42], based on which some variants have also been proposed, such as COREG [43], Tri-training [44], Multi-train [45], and CoForest [46]. As an instantiation of self-training, a semi-supervised support vector regression model was proposed and verified using 30 regression datasets and an industrial semiconductor manufacturing dataset [47]. In the method, the label distribution of the unlabeled data is estimated with two probabilistic local reconstruction models, thus providing the labeling confidence. Differently, based on the co-training paradigm, Bao et al. [48] proposed a co-training partial least squares (PLS) method for semi-supervised soft sensor development, by splitting the total process variables into two different parts, serving as two views. Instead of using single-output regression techniques, four semi-supervised multiple-output learning soft sensor models [49] were developed. In addition, by applying a spatial view, a temporal view, and a transformed view together, a multi-view transfer semi-supervised regression was developed, by combining transfer learning and co-training for air quality prediction [50]. Another example is a co-training style semi-supervised artificial neural network model for thermal conductivity prediction based on a disagreement-based semi-supervised learning principle [51]. The core of this method is constructing two artificial neural networks learners with different architectures, to label the unlabeled samples. Despite the simplicity and flexibility of implementation, the success of self-labeled SSL soft sensors heavily depends on the reliable confidence estimation of pseudo-labeled data.

According to the type of unlabeled data utilized, the above-mentioned semi-supervised modeling methods can be categorized into two groups: i.e., regularization embedding and pseudo labeling based approaches. The former methods aim to improve the model training, while the latter ones seek to enhance the model generalization capability by enlarging the labeled training data using high-confidence pseudo-labeled data. Compared to the regularization based SSL methods, pseudo labeling based ones have the merits of easy implementation and flexible wrapping with any base learning technique. Thus, in this work, we focus on pseudo-labeling semi-supervised soft sensor development. However, traditional pseudo-labeling SSL methods often encounter several drawbacks. One particular problem is the difficulty in defining confidence evaluation criterion. The most commonly used strategy is to evaluate the improvement rate of prediction performance after introducing the pseudo-labeled data [43]. Unfortunately, using such criterion cannot effectively characterize the complex hypothesis behind the labeled and unlabeled data. Another problem is that traditional pseudo-labeling strategies, such as self-training and co-training are prone to resulting in error propagation and accumulation along with the iterative learning. Thus, it is appealing to investigate the new paradigm of pseudo-labeling, to achieve reliable pseudo-label estimation.

In addition to augmenting the labeled training data, ensemble learning also plays a crucial role in enhancing the prediction performance of soft sensors [52,53,54,55]. Usually, ensemble methods can produce a strong ensemble model that is significantly more accurate than a single learner [56]. Although semi-supervised and ensemble learning approaches are two distinctive learning paradigms, they can be complementary to each other with efficient combination. On the one hand, the introduction of ensemble learning into SSL allows avoiding the difficulty in selecting model parameters and overcoming the drawback of single predictive model; thus, reducing the modeling uncertainty and improving the prediction reliability. On the other hand, the utilization of unlabeled data is helpful for improving the accuracy and diversity of base models, which is of great importance for constructing high-performance ensemble models. Therefore, the effective combination of semi-supervised and ensemble learning has attracted much attention from researchers on soft sensor development. For example, some research attempted to combine multiple homogeneous or heterogeneous base learners to construct semi-supervised ensemble models within the self-labeled learning frameworks (e.g., self-training and co-training) [50,51,57]. In addition, some scholars used unlabeled data to enhance the diversity of base models, thus improving the ensemble performance [29,58,59]. Moreover, Shao and Tian [60] proposed a soft sensor method based on a semi-supervised selective ensemble learning strategy. This method uses abundant unlabeled samples to help achieve reliable process state partition. These examples show that the combination of semi-supervised and ensemble learning is expected to improve the performance of a soft sensor model.

In light of the above-mentioned problems, in this work, a new semi-supervised soft sensor modeling method namely EnSSNCLELM was developed for high-quality prediction of industrial processes where the labeled data are limited but the unlabeled data are rich. The feasibility and superiority of the proposed method have been validated through application to a simulated fed-batch penicillin fermentation process and an industrial chlortetracycline fermentation process. Overall, the main contributions of this paper are threefold:(1)An improved supervised regression model NCLELM was developed and serves as the base learning technique for the proposed EnSSNCLELM modeling framework. Despite the fast training speed, ELM is prone to deliver unstable predictions, due to the random assignments of input weights and biases. By introducing the ensemble strategy of negative correlation learning into ELM, the NCLELM algorithm allows explicitly increasing the diversity among the base ELM models and, thus, enhancing the prediction accuracy and reliability.(2)A multi-learner pseudo-labeling optimization (MLPLO) approach is proposed to achieve pseudo-label estimation. Differently from traditional self-labeling techniques such as self-training and co-training, the MLPLO method attempts to build an explicit optimization problem with the unknown labels of the unlabeled data as the decision variables. Meanwhile, by exploring the inherent connections between labeled and unlabeled data, the individual and collaborative prediction performance of multiple learners are defined and integrated as the optimization objective. Then, an evolutionary optimization approach is adopted to solve the formulated pseudo-labeling optimization problem (PLOP), so as to obtain high-confidence pseudo-labeled samples for expanding the labeled set. A significant advantage of MLPLO is its strong capability for avoiding the error propagation and accumulation found with commonly used iterative learning.(3)By effectively combining a MLPLO strategy with ensemble modeling, the proposed EnSSNCLELM soft sensor method allows achieving the complementary advantages of semi-supervised and ensemble learning. On the one hand, semi-supervised learning is helpful for enhancing the accuracy and diversity of ensemble members by providing different high-confidence pseudo-labeled sets. On the other hand, combining multiple semi-supervised models using ensemble methods makes it possible to fully utilize the information of unlabeled data and reduce the modeling uncertainty caused by sub-optimal parameter settings and data selection.

The rest of the paper is organized as follows. Section 2 provides a brief introduction of ELM modeling and negative correlation learning (NCL). The proposed NCLELM and EnSSNCLELM soft sensor methods are elaborated in Section 3. Section 4 reports two case studies, to demonstrate the feasibility and efficiency of the proposed approach. Finally, conclusions are drawn in Section 5.

## 2. Preliminaries

### 2.1. Extreme Learning Machine

ELM [61] is a computationally efficient learning method for training single-hidden layer feedback networks (SLFNs). Unlike traditional gradient-based iterative learning, the input weights and hidden node biases of a ELM model are randomly assigned, and then the output weights of SLFNs can be obtained analytically, which enables extremely fast training of ELM.

Given a training set D={X,y} with $X={\left\{x_{i}\right\}}_{i=1}^{L}$ and $y^{l}={\left\{y_{i}\right\}}_{i=1}^{L}$ representing input and output data, respectively, where L is the number of training samples, and $x_{i}\in R^{d}$**,**
$y_{i}\in R^{1}$. Then, a standard ELM model with $N_{\mathrm{node}}$ hidden nodes and the activation function g(·) can be mathematically modeled as
(1)$$\sum_{i=1}^{N_{\mathrm{node}}}\beta_{i}g_{i}\left(x_{j}\right)=\sum_{i=1}^{N_{\mathrm{node}}}\beta_{i}g\left(w_{i}\cdot x_{j}+b_{i}\right)=o_{j},\;j=1,\cdot \cdot \cdot ,L$$
where $w_{i}={\left[w_{i,1},\;w_{i,2},\ldots ,\;w_{i,d}\right]}^{T}$ is the input weight vector that connects the *i*th hidden node and the input nodes, $\beta_{i}$ denotes the output weight that connects the *i*th hidden node and the output node, and $b_{i}$ represents the bias of the *i*th hidden node. While numerous activation functions have been defined, a simple sigmoidal function g(x)=1/(1+exp(−x)) is adopted in this study.

To train an ELM model, the fitting errors on the training data are expected to be zeros. In other words, the parameters, i.e., $\beta_{i}$, $w_{i}$, and $b_{i}$, must satisfy the following equation:(2)Hβ=y
where
(3)$$H={\left[\begin{array}{ccc} g\left(w_{1}\cdot x_{1}+b_{1}\right) & \cdots & g\left(w_{N_{\mathrm{node}}}\cdot x_{1}+b_{N_{\mathrm{node}}}\right)\\ \vdots & \ddots & \vdots \\ g\left(w_{1}\cdot x_{L}+b_{1}\right) & \cdots & g\left(w_{N_{\mathrm{node}}}\cdot x_{L}+b_{N_{\mathrm{node}}}\right) \end{array}\right]}_{L\times N_{\mathrm{node}}}$$
(4)$$\beta ={\left[\beta_{1},\cdot \cdot \cdot ,\beta_{N_{\mathrm{node}}}\right]}^{T},y={\left[y_{1},\cdot \cdot \cdot ,y_{L}\right]}^{T}$$

Since the input weights and the hidden layer biases have been randomly generated, training an ELM model is equivalent to finding a least-squares solution $\hat{\beta }$ for the linear system Hβ=y:(5)$$\hat{\beta }=H^{†}y$$
where $H^{†}$ is the Moore–Penrose generalized inverse of the hidden layer output matrix H.

### 2.2. Negative Correlation Learning

NCL [62] is a learning approach originally proposed for neural network ensembles. It introduces a correlation penalty term into the cost function of each individual network in the ensemble, thus allowing explicitly maximizing the diversity among the base models. In this way, NCL can produce individual networks interactively on the same training set and the training errors of these models tend to be negatively correlated. Given a training set $\;{\left\{x_{i},\;y_{i}\right\}}_{i=1}^{L}$, NCL combines N neural networks $f_{n}\left(x_{i}\right)$ to form an ensemble model:(6)$$f_{\mathrm{ens}}\left(x_{i}\right)=\frac{1}{N}\overset{N}{\underset{n=1}{\sum }}f_{n}\left(x_{i}\right)$$

To train network $f_{n}$, the cost function $e_{n}$ for network n is defined by:(7)$$e_{n}=\frac{1}{L}\overset{L}{\underset{i=1}{\sum }}{\left(f_{n}\left(x_{i}\right)-y_{i}\right)}^{2}+\lambda \frac{1}{L}p_{n}$$
where λ≥0 is a weighting parameter on the penalty term $p_{n}$:(8)$$\begin{array}{c} p_{n}=\sum_{i=1}^{L}\left\{\left(f_{n}\left(x_{i}\right)-f_{\mathrm{ens}}\left(x_{i}\right)\right)\cdot \sum_{m\neq n}\left(f_{m}\left(x_{i}\right)-f_{\mathrm{ens}}\left(x_{i}\right)\right)\right\}\\ =-\sum_{i=1}^{L}{\left(f_{n}\left(x_{i}\right)-f_{\mathrm{ens}}\left(x_{i}\right)\right)}^{2} \end{array}$$
where the first term is the empirical training error of the network, while the second is a correlation penalty term.

During the training process, as shown in Equations (7) and (8), all the individual networks interact with each other through their penalty terms in the cost function. Each network aims to minimize the difference between $f_{n}\left(x_{i}\right)$ and $y_{i}$, whereas the difference between $f_{\mathrm{ens}}\left(x_{i}\right)$ and $f_{n}\left(x_{i}\right)$ is maximized. In other words, NCL considers training errors from all other networks while training a network. The parameter λ controls the trade-off between the training error term and the penalty term. In the extreme case of λ=0, all individual networks are trained independently. With the increase of λ, more and more attention is paid to minimizing the correlation-based penalty.

## 3. Proposed NCLELM and EnSSNCLELM Soft Sensor Methods

To handle the issue of label scarcity for quality prediction in the industrial process, this work aims to leverage both labeled and unlabeled data, to improve the inferential performance of soft sensors, by combining semi-supervised and ensemble learning. For this purpose, we propose NCLELM and EnSSNCLELM soft sensor methods, which are described in detail in the following sections.

### 3.1. NCLELM

In recent years, ELM has gained growing popularity in soft sensor applications, due to its fast learning speed and good generalization performance [63,64,65,66]. However, ELM often produces unstable predictions, due to the uncertainties caused by the random assignments of input weights and biases in the learning process. A popular approach to this problem is to utilize an ensemble strategy to improve the stability and accuracy of the ELM model.

Since the NCL learning approach has been proven effective for enhancing the ensemble performance by explicitly maximizing the diversity among base models [67,68,69], we derived the NCLELM algorithm by introducing NCL into ELM learning. NCLELM aims to encourage the diversity among component ELM models, by reducing the correlation among their outputs, and, thus, gaining an overall good ensemble accuracy. Thanks to the fast training speed of ELM, NCLELM can also be trained very efficiently, using the multiple ELM models that are included in the learning. Subsequently, we present the proposed NCLELM method in detail.

Given a training set $D={\left\{x_{i},y_{i}\right\}}_{i=1}^{L}$ and an ensemble of ELM models containing $N_{\mathrm{ELM}}$ individuals, the error function of the *n*th individual is expressed as follows:(9)$$e_{n}\left(\beta_{n}\right)=\frac{1}{2}\left\{∥H_{n}\beta_{n}-y{∥}^{2}-\lambda ∥H_{n}\beta_{n}-f_{\mathrm{ens}}{∥}^{2}\right\}$$
where $H_{n}$ and $\beta_{n}$ denote the hidden-layer output matrix and output-layer weight vector for the *n*th individual ELM model, respectively, and $y={\left[y_{1},y_{2},\ldots y_{L}\right]}^{T}$ is a column vector. λ is a trade-off parameter on the correlation penalty term. $f_{\mathrm{ens}}$ denotes the simple average of outputs from $N_{\mathrm{ELM}}$ individuals:(10)$$f_{\mathrm{ens}}=\frac{1}{N_{\mathrm{ELM}}}\overset{N_{\mathrm{ELM}}}{\underset{n=1}{\sum }}H_{n}\beta_{n}\;$$

Then, the solution to the quadratic optimization problem in Equation (9) can be obtained by setting $\left(\partial e_{n}/\left(\partial \beta_{n}\right)\right)=0$, that is
(11)$$\frac{\partial e_{n}}{\partial \beta_{n}}={\left(H_{n}\beta_{n}-y\right)}^{T}H_{n}-\lambda \left(H_{n}\beta_{n}-f_{\mathrm{ens}}\right)\left(1-\frac{1}{N_{\mathrm{ELM}}}\right)H_{n}=0$$

By substituting Equation (10) into Equation. (11), we can get
(12)$$\frac{\partial e_{n}}{\partial \beta_{n}}={\left(H_{n}\beta_{n}-y\right)}^{T}H_{n}-\lambda \left(H_{n}\beta_{n}-\frac{1}{N_{\mathrm{ELM}}}\Sigma_{n=1}^{N_{\mathrm{ELM}}}H_{n}\beta_{n}\right)\left(1-\frac{1}{N_{\mathrm{ELM}}}\right)H_{n}=0\;$$

Furthermore, Equation (12) is transformed as
(13)$$\;[1-{\left(\frac{N_{\mathrm{ELM}}-1}{N_{\mathrm{ELM}}}\right)}^{2}\lambda ]H_{n}^{T}H_{n}\beta_{n}+\frac{N_{\mathrm{ELM}}-1}{N_{\mathrm{ELM}}^{2}}\lambda \overset{N_{\mathrm{ELM}}}{\underset{m\neq n}{\sum }}H_{n}^{T}H_{m}\beta_{m}=H_{n}^{T}y\;\;\;\;\;\;\;$$

By applying Equation (13) to all individual errors $e_{n}$ and all output weights $\beta_{n}$, and let
(14)$$G_{n,n}=[1-{\left(\frac{N_{\mathrm{ELM}}-1}{N_{\mathrm{ELM}}}\right)}^{2}\lambda ]H_{n}^{T}H_{n}$$
and
(15)$$G_{n,m}=\frac{N_{\mathrm{ELM}}-1}{N_{\mathrm{ELM}}^{2}}\lambda \overset{N_{\mathrm{ELM}}}{\underset{m\neq n}{\sum }}H_{n}^{T}H_{m}$$

Then the overall optimization of all individual ELM models is achieved by solving a linear system, as follows:(16)$$G\beta_{\mathrm{ens}}=T$$
where $G=\left[\begin{array}{ccc} G_{1,1} & \cdots & G_{1,N_{\mathrm{ELM}}}\\ \vdots & \ddots & \vdots \\ G_{N_{\mathrm{ELM}},1} & \cdots & G_{N_{\mathrm{ELM}},N_{\mathrm{ELM}}} \end{array}\right]$, $\beta_{\mathrm{ens}}=\left[\begin{array}{c} \beta_{1}\\ \vdots \\ \beta_{N_{\mathrm{ELM}}} \end{array}\right]$, and $T=\left[\begin{array}{c} H_{1}^{T}y\\ \vdots \\ H_{N_{\mathrm{ELM}}}^{T}y \end{array}\right]$.

Finally, the solution to $\beta_{\mathrm{ens}}$ can be analytically obtained as
(17)$${\hat{\beta }}_{\mathrm{ens}}=G^{-1}T\;\;\;\;\;\;\;\;\;\;\;\;\;\;\;\;\;\;\;\;\;\;\;\;\;\;\;\;\;\;\;\;\;\;\;\;\;\;\;\;\;\;\;\;\;\;\;\;\;\;\;\;\;\;\;\;\;\;$$

### 3.2. EnSSNCLELM

The proposed EnSSNCLELM method can be split into three steps: (i) obtaining high-confidence pseudo-labeled data, (ii) building diverse semi-supervised models, and (iii) combing diverse semi-supervised models.

#### 3.2.1. Formulating the Pseudo-Labeling Optimization Problem

Traditionally, the pseudo labels for the unlabeled data are estimated through self-labeling techniques, such as self-training and co-training. However, such strategies are essentially an implicit optimization, and they often encounter error accumulation and propagation. In addition, it is difficult to evaluate the confidence of pseudo labels effectively. Thus, in our recent study [70], a single-learner pseudo-labeling optimization (SLPLO) scheme was proposed, to achieve reliable estimations of pseudo labels in an explicit way. Nevertheless, one particular drawback of this approach is that only utilizing one single learner to evaluate the goodness of pseudo labels may lead to a high risk. By contrast, multiple learners can provide diverse views and insights, which are preferable to obtain a reliable confidence evaluation.

Therefore, in this work, we propose a multi-learner pseudo-labeling optimization (MLPLO) approach, aiming to produce more reliable pseudo labels than the SLPLO scheme. The task of MLPLO is to explicitly approximate the unknown labels for given unlabeled data, by relying on a group of diverse learners to explore the available information behind the labeled and unlabeled data.

As preliminary steps, the decision variables, optimization objectives, and constraints should be determined in advance; thus, formulating the pseudo-labeling optimization problem. Given a labeled training set $D^{l}=\{X^{l},y^{l}\}$, where $X^{l}={\left\{x_{i}^{l}\right\}}_{i=1}^{L}$ and $y^{l}={\left\{y_{i}^{l}\right\}}_{i=1}^{L}$ are the input and output data, respectively, and L represents the number of labeled samples. Meanwhile, denote $X^{u}={\left\{x_{i}^{u}\right\}}_{i=1}^{U}$ as the input data for the unlabeled dataset $D^{u}$, where U is the number of unlabeled samples. In addition, let $y^{u}={\left\{y_{i}^{u}\right\}}_{i=1}^{U}$ be the pseudo labels of $X^{u}$ and $D^{pl}=\left\{X^{u},y^{u}\right\}$ be the pseudo-labeled set, where $y^{u}$ are actually unknown in practical applications and remain to be estimated. Hence, $y^{u}$ is determined as the decision vector for the proposed MLPLO problem (MLPLOP).

Subsequently, the objective function should be defined to evaluate the quality of the candidate solutions. For the MLPLO approach, it is necessary to build multiple NCLELM models with high diversity, when considering NCLELM as the base learning technique. This can be achieved by repeating the random generation of input weights and biases, thereby producing diverse NCLELM models $H=\left\{{\mathrm{NCLELM}}_{1},\cdot \cdot \cdot ,{\mathrm{NCLELM}}_{M}\right\}$. It is worth noting that, during the MLPLO process, the initial configurations of input weights and biases will remain unchanged, though the output weights will be updated with the change of training set.

Within the multi-learner PLO scenario, the key to defining appropriate optimization objective lies in analyzing the individual and collaborative prediction capability of base models on the labeled and pseudo-labeled data. Specifically, four evaluation criteria are defined; including the individual accuracy using the pseudo-labeled data, individual accuracy improvement after including the pseudo-labeled data, smoothness of labeled and pseudo-labeled data, and ensemble accuracy of multiple learners using the pseudo-labeled data.

Suppose there is a pseudo-labeled set $D^{pl}=\left\{X^{u},y^{u}\right\}$ that needs to be optimized, we randomly divide $D^{pl}$ into M equal-sized training subsets $D^{pl}=\left\{D_{1}^{pl},\cdots ,D_{M}^{pl}\right\}$ and then feed these to the built NCLELM models, respectively, where $D_{i}^{pl}=\left\{X_{i}^{u},y_{i}^{u}\right\}=\left\{\left(x_{i,1}^{u},y_{i,1}^{u}\right),\cdots ,\left(x_{i,P}^{u},y_{i,P}^{u}\right)\right\}$ with P denoting the number of samples in the *i*th subset. The goal of such data partition is to promote the diversity of NCLELM models. Next, the defined evaluation criteria are discussed as follows.

(1)Individual accuracy using the pseudo-labeled data. It is well known that successful data-driven modeling greatly relies on the assumption that modeling data are independent and identically distributed. That is to say, good prediction performance on unseen samples can be attained only when the test and training data come from the same distribution. This assumption usually applies to developing data-based models for industrial processes. Thus, in the context of semi-supervised modeling, we assume that the labeled and unlabeled data are drawn from the same distribution. Intuitively, this implies that a NCLELM model trained with the pseudo-labeled data can also provide accurate predictions on the labeled set, if the pseudo labels are estimated well enough. Specifically, suppose $H^{pl}=\left\{h_{1}^{pl},h_{2}^{pl},\cdots ,h_{M}^{pl}\right\}$ denote the diverse NCLELM models learned from the pseudo-labeled subsets $D_{1}^{pl},D_{2}^{pl},\cdots ,D_{M}^{pl}$, respectively. It is obvious that we expect the performance of $h_{i}^{pl}$ on $D^{l}$ to be good if the acquired $D_{i}^{pl}$ has high quality. Furthermore, by considering all individual accuracies simultaneously, the overall accuracy is minimized:
(18)$$\underset{y^{u}}{\min}\;F\left(y^{u},X^{u},y^{l},X^{l},H\right)=\overset{M}{\underset{i=1}{\sum }}\frac{1}{L}\overset{L}{\underset{j=1}{\sum }}{\left({\hat{y}}_{i,j}^{pl}-y_{j}^{l}\right)}^{2}$$ where ${\hat{y}}_{i,j}^{pl}$ represents the predicted label of the jth labeled sample using $h_{i}^{pl}$, and $y_{j}^{l}$ is the *j*th actual label.

(2)Individual accuracy improvement after including the pseudo-labeled data. In many self-labeling semi-supervised learning algorithms [48,49,50], the pseudo labels are usually estimated from the already built predictive models. Then, the confidence of these pseudo-labeled data is evaluated according to the prediction accuracy enhancement of the model after adding the target pseudo-labeled data to the original training set. The larger the performance improvement, the higher the confidence of the pseudo-labeled data. Similarly, we also employ this way to evaluate the confidence of the optimized pseudo labels. Suppose $H^{l+pl}=\left\{h_{1}^{l+pl},h_{2}^{l+pl},\cdots ,h_{M}^{l+pl}\right\}$ are the NCLELM models learned from $D^{l}\cup D_{1}^{pl},D^{l}\cup D_{2}^{pl},\cdots ,D^{l}\cup D_{M}^{pl}$, respectively. Then, it is desirable to minimize the prediction errors of $H^{l+pl}$ on the labeled training set:
(19)$$\underset{y^{u}}{\min}\;F\left(y^{u},X^{u},y^{l},X^{l},H\right)=\overset{M}{\underset{i=1}{\sum }}\frac{1}{L}\overset{L}{\underset{j=1}{\sum }}{\left({\hat{y}}_{i,j}^{l+pl}-y_{j}^{l}\right)}^{2}$$ where ${\hat{y}}_{i,j}^{l+pl}$ is the predicted label of the jth labeled sample using $h_{i}^{l+pl}$.

(3)Smoothness of labeled and pseudo-labeled data. The objective functions in Equations (18) and (19) focus on evaluating the characteristics of pseudo-labeled subsets, but do not consider the overall confidence of all pseudo labels. According to the smoothness assumption [16,17], similar inputs will lead to similar outputs. Obviously, this assumption should also hold true for the mixed data of labeled and pseudo-labeled data, if we can obtain high-confidence pseudo labels. A popular approach using this idea, is introducing a regularization term to the cost function in semi-supervised learning, e.g., semi-supervised ELM [29] and semi-supervised deep learning [33]. In this way, the information behind the unlabeled data can be utilized in model training, to avoid overfitting. Thus, in our proposed MLPLO approach, we introduce Laplace regularization to ensure the smoothness of the labeled and pseudo-labeled data during the optimization process. After mixing the labeled samples $D^{l}$ and pseudo-labeled samples $D^{pl}$, a graph based regularization term, called smoothness objective, is defined and expected to be minimized:
(20)$$\underset{y^{u}}{\min}F\left(y^{u},X^{u},y^{l},X^{l}\right)=y^{T}Ly\;\;$$ where y denotes the outputs of the labeled and pseudo-labeled sets, i.e., $y=[y_{1},y_{2},\cdot \cdot \cdot ,y_{L},\;y_{u,1},y_{u,2},\cdot \cdot \cdot ,y_{u,U}$], and L represents a graph Laplace matrix with (L+U)×(L+U) dimensions, which can be calculated from L=D−W. D is a diagonal matrix with the elements determined as follows:(21)$$d_{ii}=\overset{L+U}{\underset{j=1}{\sum }}w_{ij}$$
where $w_{ij}\in W$ represents the connection weight between two nodes $x_{i}$ and $x_{j}$ in the graph model. Usually, $w_{ij}$ can be calculated by
(22)$$w_{ij}=e^{-\frac{∥x_{i}-x_{j}{∥}^{2}}{2\delta^{2}}},i,j=1,2,\cdot \cdot \cdot ,\;L+U$$

(4)Ensemble accuracy of multiple learners using the pseudo-labeled data. In addition to the smoothness objective, the collaborative confidence evaluation can also be achieved through the ensemble prediction performance of NCLELM models $H^{pl}=\left\{h_{1}^{pl},h_{2}^{pl},\cdots ,h_{M}^{pl}\right\}$. Thus, we expect to minimize the ensemble prediction errors of $H^{pl}$ on the labeled training set:
(23)$$\underset{y^{u}}{\min}\;F\left(y^{u},X^{u},y^{l},X^{l},H\right)=\frac{1}{L}\overset{L}{\underset{j=1}{\sum }}{\left({\hat{y}}_{j}^{\mathrm{ens}}-y_{j}^{l}\right)}^{2}$$ where ${\hat{y}}_{j}^{\mathrm{ens}}$ is the ensemble prediction output of the jth labeled sample. By ensuring the ensemble prediction accuracy, the confidence of the pseudo labels can be further improved. It should be noted that, although many strategies can be used for achieving this combination, the simple averaging rule was chosen because complex schemes are likely to cause overfitting [71].

Combining the above four optimization objectives, we obtain the synthesized objective function for the MLPLO approach, to get high-confidence pseudo labels:(24)$$\left\{ \begin{array}{cc} y^{u^{*}}=\underset{y^{u}}{\min}F\left(y^{u},X^{u},y^{l},X^{l},H\right) & =\sum_{i=1}^{M}\frac{1}{L}\sum_{j=1}^{L}{\left({\hat{y}}_{i,j}^{pl}-y_{j}^{l}\right)}^{2}+\gamma_{1}*\sum_{i=1}^{M}\frac{1}{L}\sum_{j=1}^{L}{\left({\hat{y}}_{i,j}^{l+pl}-y_{j}^{l}\right)}^{2}\\ & \;+\gamma_{2}*y^{T}Ly+\gamma_{3}*\frac{1}{L}\sum_{j=1}^{L}{\left({\hat{y}}_{j}^{\mathrm{ens}}-y_{j}^{l}\right)}^{2}\\ & s.t.y_{\min}\leq y^{u}\leq y_{\max} \end{array}\right.$$
where $y^{u}=\left[y_{1,1}^{u},y_{1,2}^{u},\cdot \cdot \cdot ,y_{1,P}^{u},\cdot \cdot \cdot ,y_{M,1}^{u},y_{M,2}^{u},\cdot \cdot \cdot ,y_{M,P}^{u}\right]$ denotes the decision vector, $\gamma_{i}\geq 0\;$( i=1,2,3) are trade-off parameters. $\left[y_{\min},y_{\max}\right]$ are the lower and upper bounds with each pair of elements for one decision variable being denoted as $\left[y_{\min,i,j},y_{\max,i,j}\right]$, where i=1,2,···,M and j=1,2,···,P. Note that the upper and lower settings can severely affect the efficiency of the MLPLO approach, so they should be determined carefully. To this end, the prediction uncertainty of the pseudo labels is first estimated through a probabilistic modeling technique and then utilized to aid in determining the boundary of the decision variables. Specifically, in this work, a GPR modeling technique [72] is applied to provide the confidence intervals of the pseudo labels, which includes three steps, i.e., first, building a GPR model from the labeled data $D^{l}$, then, obtaining the prediction variances of the unlabeled data using the GPR model, and, finally, determining the search ranges based on the confidence intervals.

#### 3.2.2. Solving the Pseudo-Labeling Optimization Problem

For the formulated MLPLO problem shown in Equation (24), it can be seen that NCLELM training and prediction are performed repeatedly during the calculation of the objective function, which means the objective function fails to satisfy continuity, differentiability, convexity, etc. Obviously, the classical optimization techniques are not suitable for solving such a optimization problem. Fortunately, evolutionary algorithms, inspired by biological phenomena, have earned great success in both machine learning and engineering applications for many tasks, such as feature selection, model selection, and optimal scheduling [73,74,75,76,77]. Unlike the classical optimization methods, evolutionary approaches conduct a parallel search, and they have the advantages of adapting to complex problems, where derivability and convexity are not required. Despite the availability of numerous evolutionary algorithms, for the sake of simplicity, one of the most well-known and commonly used approaches, genetic algorithms (GA), is chosen as an example to illustrate how to solve the MLPLOP problem through an evolutionary approach.

The strategy of producing high-confidence pseudo labels using GA optimization is illustrated in Figure 1. The details of individual representation, fitness evaluation, and evolutionary operations are presented, as follows:

(1) Individual representation. In most cases, the quality variables in the industrial process are real numbers, thus the decision variables $y^{u}=\left\{y_{1,1}^{u},y_{1,2}^{u},\cdot \cdot \cdot ,y_{1,P}^{u},\cdot \cdot \cdot ,y_{M,1}^{u},y_{M,2}^{u},\cdot \cdot \cdot ,y_{M,P}^{u}\right\}$ are encoded as chromosomes by real-number coding, as illustrated in Fig. 2. In addition, an initial population with $N_{\mathrm{pop}}$ individuals is generated randomly within the range of $\left[y_{\min},y_{\max}\right]$. 

(2) Fitness evaluation. The goal of this step is to determine the quality of candidate solutions in the populations. In this case, we compute the value of the objective function defined in Equation (24) and use its reciprocal as the fitness value to evaluate the goodness of the chromosomes.

(3) Evolutionary operations. If the stopping condition is not satisfied, the GA needs to generate an offspring population by executing evolutionary operations; i.e., selection, crossover, and mutation. The selection operator can find the good solutions with the largest fitness values in a population. Crossover creates new individuals, by combing the genes of one individual with those of another. In mutation, the genes in individuals will make small random changes to create mutation children. Such an operation provides genetic diversity and allows enlarging the search space.

After performing the GA based MLPLO optimization, the optimal solution encoding the best pseudo labels is obtained, i.e., $y_{*}^{u}=\left[y_{*,1,1}^{u},y_{*,1,2}^{u},\cdot \cdot \cdot ,y_{*,1,P}^{u},\cdot \cdot \cdot ,y_{*,M,1}^{u},y_{*,M,2}^{u},\cdot \cdot \cdot ,y_{*,M,P}^{u}\right]$. Then, by combining $y^{u^{*}}$ with its corresponding $X^{u}$, we get the optimized pseudo-labeled set $D^{pl}=\left\{X^{u},y^{u^{*}}\right\}$.

#### 3.2.3. Building Diverse SSNCLEM Base Models

In order to obtain a good ensemble model, we need to construct accurate and diverse base models. Once the MLPLO process is completed, a group of pseudo-labeled samples can be obtained and further combined with the labeled set to build a set of SSNCLELM models. It is fairly easy to see that the performance of the SSNCLELM model is highly dependent on the quality of the optimized pseudo-labeled samples. Hence, to ensure the reliability of the MLPLO process, the following two problems should be handled properly:

(1) Selection of the unlabeled samples. Though there are a large number of unlabeled data available for SSL learning in practical industrial processes, the number of unlabeled samples for the MLPLO approach cannot be too large. As indicated in Figure 2, the dimension of the decision variables is equal to the size of the selected unlabeled data; hence, inclusion of too many unlabeled samples will inevitably make the MLPLOP problem become a large-scale optimization problem [78]. In this case, it is very difficult to obtain good a search performance, efficiency, and effectiveness. In addition, introducing too many pseudo-labeled samples into the SSNCLELM model construction will also weaken the influence of the labeled samples, thus leading to high modeling risk. Therefore, it is more realistic to generate diverse small-scale pseudo-labeled sets and then construct diverse SSNCLELM base models.

(2) Setting of the trade-off parameters $\{\gamma_{1}$,$\;\gamma_{2}$,$\;\gamma_{3}\}$. Besides the unlabeled data selection, another important factor affecting the optimization performance of the MLPLO approach, is determining an appropriate combination of trade-off parameters. In practice, it is a difficult task to find a globally optimal combination of the three parameters, and there is usually more than one combination that can meet the requirements. Considering this, a natural method is to apply diverse settings of trade-off parameters for the MLPLO process, thus avoiding the difficulties in parameter selection, as well generating diverse pseudo-labeled samples. This is very helpful for enhancing the diversity of SSNCLELM base models.

In light of the above problems, we aim to generate diverse SSNCLELM models through a multi-model perturbation mechanism, which combines random assignment of NCLELM model parameters, random selection of small-scale unlabeled samples for MLPLO optimization, and diverse settings of trade-off parameters $\left\{\gamma_{1},\gamma_{2},\gamma_{3}\right\}$.

Let $\Phi_{k}={\left\{{\left\{{\mathrm{NCLELM}}_{\mathrm{init},i}\right\}}_{i=1}^{M},D^{u^{\prime }},\;{\left\{\gamma_{i}\right\}}_{i=1}^{3}\right\}}_{k}$ denote the specialized setting for the *k*th run of diverse SSNCLELM model construction, where ${\left\{{\mathrm{NCLELM}}_{\mathrm{init},i}\right\}}_{i=1}^{M}$ are generated by randomly assigning input weights and biases, the selected unlabeled data $D^{u^{\prime }}$ are randomly resampled from $D^{u}$, and ${\left\{\gamma_{i}\right\}}_{i=1}^{3}$ is a combination from $\gamma_{1}\in {\left\{\gamma_{1,i}\right\}}_{i=1}^{k_{1}}$, $\gamma_{2}\in {\left\{\gamma_{2,i}\right\}}_{i=1}^{k_{2}}$, and $\gamma_{3}\in {\left\{\gamma_{3,i}\right\}}_{i=1}^{k_{3}}$. By applying $\Phi_{k}$ to the proposed MLPLO approach, an enlarged labeled set $D^{l+pl}=D^{l}\cup D_{k}^{pl}$ can be obtained and further used to update the initial models ${\left\{{\mathrm{NCLELM}}_{\mathrm{init},i}\right\}}_{i=1}^{M}$, thus resulting in a set of diverse SSNCLELM models $\left\{{\mathrm{SSNCLELM}}_{k,1},{\mathrm{SSNCLELM}}_{k,2},\ldots ,{\mathrm{SSNCLELM}}_{k,M}\right\}$. By repeating the above procedure $K=k_{1}\times k_{2}\times k_{3}$ times, we can obtain a total of K×M diverse SSNCLELM models:(25)$$\{\begin{array}{c} \Phi_{1}:\left\{{\mathrm{SSNCLELM}}_{1,1},{\mathrm{SSNCLELM}}_{1,2},\ldots ,{\mathrm{SSNCLELM}}_{1,M}\right\}\\ \begin{array}{c} \vdots \end{array}\\ \Phi_{K}:\left\{{\mathrm{SSNCLELM}}_{K,1},{\mathrm{SSNCLELM}}_{K,2},\ldots ,{\mathrm{SSNCLELM}}_{K,M}\right\} \end{array}\;\;\;\;\;\;\;\;\;\;\;\;\;\;\;\;\;\;\;\;\;\;\;\;\;\;$$

#### 3.2.4. Combining Diverse SSNCLEM Base Models

After obtaining diverse SSNCLELM models, we need to combine these models in an appropriate manner. It is a common practice to utilize all base models for the ensemble construction. However, according to Zhou’s the finding of ‘many could be better than all’ [71], it may be better to combine only part rather than all of the base models. The underlying reason is that there may exist poor base models, which can hurt the ensemble performance instead of improving it. Thus, ensemble pruning has been strongly emphasized for ensemble model construction [60,79,80]. Regarding the area of semi-supervised learning, the introduction of unlabeled and pseudo-labeled information may lead to unsafe semi-supervised learning [81]. That is, the prediction capability of a model cannot be improved, but encounters deterioration after including the unlabeled and pseudo-labeled data.

Thus, we propose an ensemble pruning strategy based on performance improvement evaluation. First, the supervised prediction performance of a NCLELM model learnt from the labeled data is evaluated on an independent validation set $D_{val}=\left\{X_{val},y_{val}\right\}$. Then, a SSNCLELM model is trained from the augmented labeled set containing the pseudo-labeled data, and its performance is further evaluated using the same validation set. Finally, the performance improvement ratio is calculated, to decide whether the candidate SSNCLELM model is retained or not. Specifically, for the *i*th SSNCLELM, the performance improvement ratio (PIR) is calculated as follows:(26)$$RMSE_{i}^{init}={\mathrm{NCLELM}}_{i}.\mathrm{evaluate}\left(D_{val}\right)$$
(27)$$RMSE_{i}={\mathrm{SSNCLELM}}_{i}.\mathrm{evaluate}\left(D_{val}\right)$$
(28)$$PIR_{i}=(RMSE_{i}^{init}-RMSE_{i})/\;RMSE_{i}^{init}$$
where $RMSE_{i}^{init}$ represents the RMSE of the *i*th NCLELM on $D_{val}$ and $RMSE_{i}$ is the RMSE of the *i*th SSNCLELM on $D_{val}$. Then, a threshold $PIR_{th}$ is required to select SSNCLELM models whose PIR values exceed $PIR_{th}$. In general, $PIR_{th}$ can be set to 0, as used in our case studies, which means that those SSNCLELM models without performance improvement on the validation set are discarded. 

After the ensemble pruning, suppose that a total of S≤K×M SSNCLELM models are retained for ensemble construction. Since the simple average rule cannot always function well, it is appealing to integrate the base models using a weighting scheme. Hence, in this work, a stacking strategy is employed for model combination. The basic idea of this approach is to use the built SSNCLELM models as the first-level learners, and then train the second-level learner by using the outputs of the first-level leaners as inputs, while the original output is still regarded as the output of the model. That is,
(29)$$\mathrm{EnSSNCLELM}=\mathrm{Stacking}\left({\mathrm{SSNCLELM}}_{sel,1},\cdots ,{\mathrm{SSNCLELM}}_{sel,S}\right)$$

Specifically, given an independent validation set $D_{val}=\left\{X_{val},y_{val}\right\}$, the prediction outputs ${\hat{y}}_{val}=\left\{{\hat{y}}_{val}^{1},\;{\hat{y}}_{val}^{2},\cdot \cdot \cdot ,{\hat{y}}_{val}^{S}\right\}$ on $D_{val}$ can be obtained using the selected base models ${\left\{{\mathrm{SSNCLELM}}_{sel,i}\right\}}_{i=1}^{S}$. Then, a stacking model can be built from $\left\{{\hat{y}}_{val},y_{val}\right\}$, where ${\hat{y}}_{val}$ and $y_{val}$. are used as input and output data, respectively. To effectively handle the collinearity issue among the SSNCLELM base models, a PLS stacking model is built. Thus, for one query sample $x_{q}$, the ensemble prediction output ${\hat{y}}_{q}$ can be obtained as
(30)$${\hat{y}}_{q}=\beta_{0}+\overset{S}{\underset{i=1}{\sum }}\beta_{i}{\hat{y}}_{q}^{i}$$
where $\left\{\beta_{0},\;\beta_{1},\;\cdots \;,\;\beta_{S}\right\}$ denotes the PLS regression coefficients, and ${\hat{y}}_{q}^{i}$ is the predicted value from the *i*th SSNCLELM model.

#### 3.2.5. Implementation Procedure of the EnSSNCLELM Soft Sensor

The pseudo-code of the proposed EnSSNCLELM soft sensor method is described in Algorithm 1, and its workflow is illustrated in Figure 3.
**Algorithm 1** EnSSNCLELM soft sensor method.INPUT: $D^{l}$: labeled set    $D^{u}$: unlabeled set    $D_{\mathrm{val}}$: validation set    $n_{\mathrm{pop}}$: population size for GA optimization    $n_{\mathrm{gen}}$: maximum number of iterations    $\left\{\gamma_{1},\gamma_{2},\gamma_{3}\right\}$: trade-off parameters    {$N_{\mathrm{ELM}},N_{\mathrm{node}},\lambda$}: the number of ELM in each NCLELM, the number of hidden node size of ELM base models, the tradeoff parameter of NCLELM method, respectively    U′: the number of unlabeled data $D^{u}\prime$ that requires pseudo labeling    M: the number of NCLELM models for the MLPLO approachPROCESS:1: Generate diverse combinations of parameters ${\left\{\gamma_{1,k},\gamma_{2,k},\gamma_{3,k}\right\}}_{k=1}^{K}$ for the MLPLO optimization, where $K=k_{1}\times k_{2}\times k_{3}$, $\gamma_{1}\in {\left\{\gamma_{1,i}\right\}}_{i=1}^{k_{1}}$, $\gamma_{2}\in {\left\{\gamma_{2,i}\right\}}_{i=1}^{k_{2}}$ and $\gamma_{3}\in {\left\{\gamma_{3,i}\right\}}_{i=1}^{k_{3}}$;%% Building diverse SSNCLELM models2: for k=1 to K do   %% Estimating the pseudo labels through the MLPLO approach3:    Generate M new initial NCLELM models ${\left\{{\mathrm{NCLELM}}_{\mathrm{init},i}\right\}}_{i=1}^{M}$ with the hyperparameters $\left\{N_{\mathrm{ELM}},N_{\mathrm{node}},\lambda \right\}$ by randomly setting the input weights and biases;4:   Select a small-scale unlabeled set $D^{u^{\prime }}$ by randomly resampling from $D^{u}$;5:   Determine the pseudo labels of $D^{u^{\prime }}$ as the decision variables $y_{k}^{u}=\left\{y_{1,1}^{u},y_{1,2}^{u},\cdot \cdot \cdot ,y_{1,P}^{u},\cdot \cdot \cdot ,y_{M,1}^{u},y_{M,2}^{u},\cdot \cdot \cdot \;,y_{M,P}^{u}\right\}$ with $P=U^{\prime }/M$ and set the lower and upper bounds $\left[y_{\min},y_{\max}\right]$ for $y_{k}^{u}$ based on GPR regression analysis;6:   Encode the decision variables as real-valued chromosome s, and randomly generate an initial population $Pop={\left\{s_{i}\right\}}_{i=1}^{n_{pop}}$ with $n_{pop}$ individuals within the decision boundary;7:   Repeat $n_{gen}$ times:8:    Decode the pseudo labels of $y_{k}^{u}$ from each individual chromosome in the population;9:    Evaluate the fitness of each individual $s_{i}$ in Pop according to Equation (24);10:   Generate an offspring population by performing selection, crossover, and mutation operations;     mutation operations;11:   end of Repeat12:   Select the best individual $s^{*}$ from the final population;13:   Obtain the best pseudo labels $y^{u^{*}}$ by decoding the chromosome $s^{*}$ and form the pseudo-labeled set $D_{k}^{\mathrm{pl}}$;%% Building diverse SSNCLELM models for each run14:  Update the initial models ${\left\{{\mathrm{NCLELM}}_{\mathrm{init},i}\right\}}_{i=1}^{M}$ using the optimized enlarged labeled set $D^{l}\cup D_{k}^{\mathrm{pl}}$ to obtain the semi-supervised models ${\left\{{\mathrm{SSNCLELM}}_{k,i}\right\}}_{i=1}^{M}$;15: end of for%% Ensemble pruning based on performance improvement evaluation16: Let $\Omega =\left\{{\mathrm{SSNCLELM}}_{1,1},\cdot \cdot \cdot ,{\mathrm{SSNCLELM}}_{1,M},\cdot \cdot \cdot ,{\mathrm{SSNCLELM}}_{K,1},\cdot \cdot \cdot ,{\mathrm{SSNCLELM}}_{K,M}\right\}$;17: for i=1 to K do18:  for j=1 to M do19:   ${\mathrm{RMSE}}_{i,j}={\mathrm{SSNCLELM}}_{i,j}$.evaluate($D_{\mathrm{val}}$);20:   ${\mathrm{PIR}}_{i,j}=({\mathrm{RMSE}}_{i,j}^{\mathrm{init}}-{\mathrm{RMSE}}_{i,j})/{\;\mathrm{RMSE}}_{i,j}^{\mathrm{init}}$;21:   if ${\mathrm{PIR}}_{i,j}\geq {\mathrm{PIR}}_{\mathrm{th}}$;22:    Add ${\mathrm{SSNCLELM}}_{i,j}$ to the selected model pool $\Omega_{\mathrm{sel}}$;23:   end of if24:  end of for25: end of for26: Let $\Omega_{\mathrm{sel}}=\left\{{\mathrm{SSNCLELM}}_{\mathrm{sel},1},{\mathrm{SSNCLELM}}_{\mathrm{sel},2},\ldots ,{\mathrm{SSNCLELM}}_{\mathrm{sel},S}\right\}$;%% Constructing PLS stacking ensemble model27: Let $D_{\mathrm{val}}=\left\{X_{\mathrm{val}},y_{\mathrm{val}}\right\}$;28: $\left\{{\hat{y}}_{\mathrm{val}}^{1},{\hat{y}}_{\mathrm{val}}^{2},\cdot \cdot \cdot ,{\hat{y}}_{\mathrm{val}}^{S}\right\}=\Omega .\mathrm{predict}\left(X_{\mathrm{val}}\right)$;29: Let ${\hat{y}}_{\mathrm{val}}=\left\{{\hat{y}}_{\mathrm{val}}^{1},{\hat{y}}_{\mathrm{val}}^{2},\cdot \cdot \cdot ,{\hat{y}}_{\mathrm{val}}^{S}\right\}$;30: ${\left\{\beta_{m}\right\}}_{m=0}^{S}=\mathrm{PLS}.\mathrm{fit}\left({\hat{y}}_{\mathrm{val}},y_{\mathrm{val}}\right)$, where β is the regression coefficient of PLS;%% Online prediction31: Given a query sample $x_{q}$;32: ${\left\{{\hat{y}}_{q,m}\right\}}_{m=1}^{S}=\Omega .\mathrm{predict}\left(x_{q}\right)$;33: The final prediction output ${\hat{y}}_{q}$ of $x_{q}$ are obtained by Equation (30).OUTPUT: ${\hat{y}}_{q}$

## 4. Case Studies

In this section, the performance of the proposed NCLELM and EnSSNCLELM soft sensor methods are evaluated through a simulated penicillin fermentation process and an industrial fed-batch chlortetracycline (CTC) fermentation process. The methods for comparison are as follows:

(1) ELM [61]: the supervised single ELM model.

(2) NCLELM: the proposed supervised ensemble of ELM models using the NCL rule.

(3) EnNCLELM_avg_: the supervised ensemble of NCLELM models using the simple averaging rule.

(4) EnNCLELM_pls_: the supervised ensemble of NCLELM models using the PLS stacking strategy.

(5) CoELM: the co-training based semi-supervised ELM model, where the confidence evaluation method proposed by Zhou and Li (2005) [43] is used and the Euclidean distance is used to choose similar samples for confidence evaluation, and two different ELM models are trained using two randomly generated feature subsets.

(6) SSELM [29]: semi-supervised ELM model embedding the graph based Laplacian regularization.

(7) EnSSNCLELM_avg_: the semi-supervised ensemble of diverse SSNCLELM models using the simple averaging rule.

(8) EnSSNCLELM: the proposed method, the semi-supervised ensemble of SSNCLELM models using PLS stacking strategy.

To quantitatively assess the prediction performance of different soft sensors, the root-mean-square error (RMSE) and coefficient of determination ($R^{2}$) are used:(31)$$\mathrm{RMSE}=\sqrt{\frac{1}{N}\overset{N_{\mathrm{test}}}{\underset{n=1}{\sum }}{\left(y_{n}-{\hat{y}}_{n}\right)}^{2}}$$
(32)$$R^{2}=1-\frac{\sum_{n=1}^{N_{\mathrm{test}}}{\left({\hat{y}}_{n}-y_{n}\right)}^{2}}{\sum_{n=1}^{N_{\mathrm{test}}}{\left({\hat{y}}_{n}-\bar{y}\right)}^{2}}$$
where $\;N_{\mathrm{test}}$ is the number of testing samples; ${\hat{y}}_{n}$ and $y_{n}$ are the nth predicted and actual values of the testing sample, respectively; and $\bar{y}$ is the mean value of the output variable.

### 4.1. Application to Penicillin Fermentation Process

#### 4.1.1. Process Description

The penicillin fermentation process has been widely used for investigating the modeling, monitoring, and controlling of batch processes [82,83]. A flow diagram of the process is illustrated in Figure 4. Since formation of secondary metabolites (in this case, penicillin) is not associated with cell growth, the cells generally grow in a batch culture and then achieve synthesis of antibiotics through a fed-batch operation. In general, the penicillin production process lasts for 400 h, during which two cascade controllers are used to maintain the pH and temperature. In addition, the sterile substrate and air are continuously fed into the bioreactor to supply the nutrients for cell growth and product formation, as well as satisfying the oxygen consumption necessary for the microorganisms. 

In our experimental study, the process data were collected from the PenSim platform, which was developed by the Process Modeling, Monitoring, and Control Research Group of Illinois Institute of Technology for simulating the fed-batch penicillin fermentation process [84]. The software PenSim can be downloaded from the website on http://simulator.iit.edu/web/pensim/index.html (accessed on 5 August 2019). In this work, penicillin concentration was chosen as the difficult-to-measure variable, while the relevant process variables in Table 1 were used as the input variables of soft sensor models. Moreover, the sampling interval was set as 0.5 h and the default operation settings were considered. 

#### 4.1.2. Prediction Performance and Discussion of NCLELM

To verify the effectiveness and superiority of the proposed NCLELM, we conducted the experiments as follows:

(1) Two application scenarios with different labeled data sizes were considered. The first case used 200 labeled samples collected from five batches for model training, while the second case used 800 labeled samples from the same batches. In addition, 800 labeled samples from another five batches were used for testing.

(2) The prediction performance of NCLELM with respect to different parameter configurations $\left\{N_{\mathrm{ELM}},N_{\mathrm{node}},\lambda \right\}$ was analyzed, where $N_{\mathrm{ELM}}\in \left\{5,10,15,20\right\},$ $N_{\mathrm{node}}\in \left\{5,6,\cdots ,30\right\}$, and λ∈{0,0.2,⋯1}. Among these parameters, $N_{\mathrm{ELM}}$ and $N_{\mathrm{node}}$ affect the model complexity significantly, while λ indicates the degree of concern placed on minimizing the correlation-based penalty.

(3) A comparison between NCLELM and ELM was performed to demonstrate the advantage of the NCL rule. Since both methods exhibit a random nature, due to the random assignments of input weights and biases; for the sake of fairness, their prediction accuracies are provided as the simple average of those from 50 runs.

Figure 5 illustrates the testing performance of the proposed NCLELM approach for different settings of parameters $\left\{N_{\mathrm{ELM}},N_{\mathrm{node}},\lambda \right\}$, where the prediction performance of ELM is also given for comparison. The effects of hyper-parameters $\left\{N_{\mathrm{ELM}},N_{\mathrm{node}},\lambda \right\}$ on the NCLELM model performance are analyzed first. It is readily seen that, overall, the prediction errors of NCLELM decreased with the increase of $N_{\mathrm{node}}$. It is also noticeable that, when large $N_{\mathrm{node}}$ was used, the performance enhancement resulting from the increase of $N_{\mathrm{node}}$ became subtle. This was mainly because a high model complexity tends to cause overfitting when the training data are insufficient. For example, in the application case with 200 labeled training samples, as shown in Figure 5a, the testing accuracy was not improved significantly after $N_{\mathrm{node}}$ exceeded 15, due to the scarcity of the labeled training data. In comparison, as shown in Figure 5b, the trend plots from the case with 800 labeled training samples decreased more sharply and the testing RMSE values continuously became smaller when $N_{\mathrm{node}}$ increased.

Similarly to $N_{\mathrm{node}}$, $N_{\mathrm{ELM}}$ also had a great impact on the NCLELM model complexity. It can be seen from the four subplots in Figure 5a, when a small $N_{\mathrm{node}}$ was used for NCLELM, a large $N_{\mathrm{ELM}}$ was more likely to produce a high prediction accuracy. However, when a large $N_{\mathrm{node}}$ was considered, the increase of $N_{\mathrm{ELM}}$ only led to a slight enhancement of the testing accuracy. A similar phenomenon can also be found in Figure 5b. Based on the above analysis, we can conclude that, when the training data size is small, a less complex model structure is preferable; while a more complex model, which can be built by increasing $N_{\mathrm{node}}$ and/or $N_{\mathrm{ELM}}$, is required when the training data are sufficient.

Unlike $N_{\mathrm{node}}$ and $N_{\mathrm{ELM}}$, λ was introduced to control the importance of the correlation based penalty objective in the training loss function. When the individual model complexity was not so high, e.g., in the case with a small $N_{\mathrm{node}}$, a large λ was required to ensure the prediction accuracy of NCLELM. One possible reason for this is that, in such a situation, it is desirable to emphasize the NCL effect of NCLELM, to guarantee the diversity among the ELM base models, which is crucial for building a high-performance ensemble model. As the model complexity grows, the benefits from introducing the NCL rule decline, even though a large λ is employed. One possible answer to this question is that, when a complex model structure is used, the diversity among the base ELM models for the NCLELM construction can be effectively maintained only through the inherent random assignments of the NCLELM parameter configurations. In this case, the NCL rule-based diversity improvement is not obvious. In the case of λ=0, NCLELM degenerates as the simple averaging rule based ensemble of ELM models. 

Subsequently, we compared the prediction accuracy of the NCLELM and ELM methods. As can be seen from the figure, similarly to NCLELM, the prediction performance of ELM was also enhanced with the increase of $N_{\mathrm{node}}$. When the training data are limited (e.g., $N_{\mathrm{trn}}=200$), a moderate $N_{\mathrm{node}}$, such as 20 is acceptable, whereas a larger $N_{\mathrm{node}}$, such as 30, is more suitable for a case with a relatively large training data size, such as $N_{\mathrm{trn}}=800$. Meanwhile, we found that the performance of NCLELM was always much better than that of ELM, regardless of the parameter settings and training data sizes, which implies the outstanding performance of NCLELM. The above experimental results fully confirm the effectiveness and superiority of the proposed NCLELM over traditional ELM.

#### 4.1.3. Analysis and Comparison of EnSSNCLELM Prediction Results

To evaluate the effectiveness of the proposed EnSSNCLELM soft sensor method, five batches, including 4000 labeled samples, were collected for soft sensor training. To imitate the practical application scenario with scarce labeled data but rich unlabeled data, the obtained training set was further divided into two parts: a small-sized labeled training set with only 200 labeled samples, and a large-sized unlabeled training set with 3800 samples, whose actual labels had been removed. In addition, an independent validation set including 40 labeled samples was collected from two batches for ensemble pruning and stacking. Moreover, another five batches were obtained to assess the online prediction performance of the soft sensors.

To build well-performing soft sensor models, some critical parameters of different soft sensor methods should be determined in advance by cross-validation or trial and error, as follows:

(1) ELM: The hidden node size $N_{\mathrm{node}}$ was set as 20 by 5-fold cross-validation.

(2) NCLELM: The hyperparameters {$N_{\mathrm{ELM}},N_{\mathrm{node}},\lambda$} were set as {5,20,0.6} by 5-fold cross-validation, respectively.

(3) CoELM and SSELM: $N_{\mathrm{node}}$ for the two methods were selected as 20, the maximum number of iterations for CoELM was set as 300, and the penalty coefficient for SSELM was set as 0.3, by trial and error.

(4) EnSSNCLELM: The hyperparameters {$N_{\mathrm{ELM}},N_{\mathrm{node}},\lambda$} for the SSNCLELM base models were the same as those of NCLELM. The M and $U^{\prime }$ were set to 3 and 300, respectively, for formulating the MLPLOP problem. In addition, the parameter configurations for the GA based MLPLO optimization of EnSSNCLELM were set as follows: the population size $n_{pop}=50$ and the maximum number of iterations $n_{gen}=50$ for GA optimization, and diverse combinations of trade-off parameters $\left\{\gamma_{1},\gamma_{2},\gamma_{3}\right\}$ were generated from the candidate set of {0,0.01, 0.1, 0.5, 1}. In addition, the number of principal components of the PLS stacking model was determined by 5-fold cross-validation.

Considering that the EnNCLELM_avg_, EnSSNCLELM_avg_, and EnNCLELM_pls_ methods are actually the degraded versions of the proposed EnSSNCLELM approach, their parameter configurations were set to the same as those of EnSSNCLELM. Under the above parameter settings, a total of 375 SSNCLELM models were first built based on the extended labeled data. After ensemble pruning, 350 models were retained for constructing the ensemble models EnSSNCLELM_avg_ and EnSSNCLELM. Meanwhile, the NCLELM base models, corresponding to the SSNCLELM ones, were selected for EnNCLELM_avg_ and EnNCLELM_pls_ modeling.

The prediction results of different soft sensors on the testing set are tabulated in Table 2. It can be seen that ELM had the worst performance as a single model, and NCLELM delivered a better accuracy, due to the introduction of the NCL ensemble strategy. By using ELM as the base learning technique, two representative semi-supervised methods, i.e., CoELM and SSELM, were developed using co-training and embedding graph regularization, respectively. Though CoELM and SSELM include unlabeled data and, thus, achieved an accuracy improvement, their prediction performance was still very poor. Furthermore, by using NCLELM for base model building, two supervised ensemble models, namely EnNCLELM_avg_ and EnNCLELM_pls_ are developed by introducing a simple averaging rule and PLS stacking strategy, respectively. It is readily observed that, compared to NCLELM, both ensemble methods provided a better prediction accuracy, and the comparison between EnNCLELM_pls_ and EnNCLELM_avg_ further implied the superiority of the stacking scheme over to the simple averaging strategy. In addition, compared to EnNCLELM_avg_, EnSSNCLELM_avg_ obtained a much better prediction accuracy, due to the utilization of unlabeled data information, which can be also revealed by the comparison between EnNCLELM_pls_ and EnSSNCLELM. Overall, due to the efficient combination of ensemble and semi-supervised learning, the proposed EnSSNCLELM approach provided a much better prediction performance than the other traditional supervised and semi-supervised methods.

Figure 6 displays the trend plots of the predicted and actual penicillin concentrations using ELM, EnNCLELM_pls_, and EnSSNCLELM methods, to provide a more intuitive comparison of model performance. It is evident that the predictions of NCLELM have large deviations from the actual values, especially in the fermentation period of 40–150 h. In comparison, EnNCLELM_pls_ significantly reduced the deviations, due to the introduction of ensemble learning. Furthermore, we can see that EnSSNCLELM produced smaller deviations than EnNCLELM_pls_ in the period of 70–180 h, although their prediction accuracies are comparable in other zones. The superior predictions of EnSSNCLEM were mainly due to the introduction of ensemble learning and the utilization of pseudo-labeled data. On the one hand, ensemble learning allows significantly reducing the prediction uncertainty from the base models, which is especially useful for handing the extreme deviations that often occur for a single model. On the other hand, using an enlarged training set can effectively improve the prediction capability of the base models, which is helpful for constructing a good ensemble.

Subsequently, the individual model performance and ensemble prediction results from the proposed SSNCLELM approach are illustrated in Figure 7. Since, EnSSNCLELM is essentially an ensemble method, both the quality of the SSNCLELM base models and the combination scheme are critical to the ensemble construction. First, the base model performance was investigated under the supervised and semi-supervised learning settings. To this end, once a NCLELM model was built, a corresponding SSNCLELM model with the same input weights and biases was trained by the enlarged labeled set. From the figure, it can be observed that the prediction RMSE values of SSNCLELM models were much smaller than those of the NCLELM models, except for in very few cases. Obviously, such significant performance enhancement is entirely attributed to the incorporation of semi-supervised learning. Then, the ensemble prediction performance using different combination strategies was compared. With the NCLELM and SSNCLELM base models, EnNCLELM_avg_ and EnSSNCLELM_avg_ were built using the simple averaging rule, while EnNCLELM_pls_ and EnSSNCLELM employed PLS stacking. One can observe that, both EnNCLELM_avg_ and EnSSNCLELM_avg_ only obtained a prediction accuracy that was inferior to the best base model, which is mainly because the simple averaging ignores the differences among the base models. By contrast, EnNCLELM_pls_ provided a prediction accuracy that was close to that of the best base model, while our proposed EnSSNCLELM approach was greatly superior to all base models. These results further confirm the effectiveness of the proposed pseudo-labeling technique and the PLS stacking strategy.

The success of the proposed EnSSNCLELM method depends highly on the quality of the obtained pseudo labels. Thus, one remaining question is how well the pseudo labels can coincide with the actual labels. Although in real-world applications, the actual labels for the unlabeled data are unknown, in this case study, the true labels could be easily acquired from the simulated platform. Hence, as illustrated in Figure 8, we present two types of comparisons: evaluating the fitting degree of the pseudo and true labels, and comparing the testing performance, before and after introducing pseudo-labeled data. As indicated by the prediction results from the two example runs, the pseudo and the actual labels achieved a strong agreement with small prediction RMSE values. Moreover, it is noticeable that ${\mathrm{RMSE}}_{\mathrm{SSNCLELM}}$ was much lower than ${\mathrm{RMSE}}_{\mathrm{NCLELM}}$, due to the inclusion of pseudo-labeled samples. These observations further imply the effectiveness and reliability of the proposed pseudo-labeling optimization technique.

The above results show that the proposed EnSSNCLELM method can effectively combine semi-supervised learning with ensemble learning, thus allowing resolving the label scarcity issue and improving the estimation reliability of difficult-to-measure variables in an industrial process.

### 4.2. Application to an Industrial Chlortetracycline Fermentation Process

#### 4.2.1. Process Description

Chlortetracycline (CTC) is a type of broad-spectrum antibiotic, which has been widely used in pharmaceutical and agricultural production and animal husbandry. The industrial CTC fermentation process under study is carried out by Charoen Pokphand Group Co., Ltd (Zhumadian, China)., as illustrated in Figure 9 [65]. Generally, CTC is produced through the cultivation of Streptomyces aureofaciens in an air-lift stirred fermenter with a volume of 120 m^3^. During the fed-batch cultivation process, real-time measurements of substrate concentration are desirable, to facilitate efficient feeding control. However, up to now, the feeding control is often still operated manually, due to the lack of reliable online hardware sensors for substrate concentration measurement, which is usually achieved through offline chemical analysis, with a long delay of 6 h. Thus, soft sensors are desirable to enable real-time estimations of substrate concentration. In this work, the process variables listed in Table 3 were used as inputs for soft sensor development.

#### 4.2.2. Analysis and Comparison of Prediction Results

In this case study, a total of 14 batches of process data were collected from an industrial fermenter, including 351 labeled samples and 3183 unlabeled samples. Among the labeled samples, 124 were utilized for model training, 29 were used for validation, and the remaining 198 samples served as the testing set. In addition, 3183 unlabeled samples were used for the pseudo labeling optimization and semi-supervised learning. Moreover, the parameter settings for different soft sensor models were the same as those in the application to the penicillin fermentation process. Consequently, a total of 375 SSNCLELM models were obtained, and 264 were selected using the ensemble pruning strategy for the construction of the EnSSNCLELM model.

The substrate concentration prediction results for the different soft sensor methods are presented in Table 4. It is obvious that ELM performed much worse than the other methods, while the proposed EnSSNCLELM method performed best. Compared with the supervised single ELM model, NCLELM achieved a performance enhancement, due to the introduction of NCL ensemble learning. Meanwhile, both CoELM and SSELM also reduced the prediction errors, due to the use of unlabeled data information. Furthermore, by applying an ensemble learning philosophy and using NCLELM as the base learning technique, EnNCLELM_avg_ and EnNCLELM_pls_ delivered a much better estimation performance than the single NCLELM. Comparing EnNCLELM_pls_ EnNCLELM_avg_ indicates that the PLS stacking outperformed the simple averaging rule. In addition, compared with EnNCLELM_pls_, the prediction accuracy of EnSSNCLELM was significantly improved, because of the incorporation of semi-supervised learning. A similar situation also appeared in the performance comparison of EnNCLELM_avg_ and EnSSNCLELM_avg_. Similar to the application in the penicillin fermentation process, once again, the application results demonstrated the superiority of the proposed method over the traditional supervised and semi-supervised methods.

More intuitively, the scatter plots of the predicted versus actual substrate concentration values using the NCLELM, EnNCLELM_pls_, and EnSSNCLELM models are depicted in Figure 10. In the scatter plot, the accuracy of the predictive model is evaluated by the closeness of points to the diagonal line. The closer the scatter points are to the diagonal line, the higher the model accuracy. From the figure, one can see at a glance that the test samples exhibit an unbalanced distribution. In the small-value zone (about 1–4), the testing samples are dense, while in the large-value zone (about 4–7), the samples are sparse. Furthermore, it can be readily observed that the NCLELM, EnNCLELM_pls_, and EnSSNCLELM methods provided a different prediction performance. For the small-value zone, NCLELM suffers from large deviations from the diagonal line, implying a poor prediction accuracy. EnNCLELM_pls_ reduces the deviations by introducing ensemble learning, thus, indicating a significant accuracy improvement. In comparison, EnSSNCLELM achieved the best closeness to the diagonal line, due to the exploitation of unlabeled data, which led to further enhancement of the model performance. Differently, in the large-value zone, one can find an interesting aspect, in that EnNCLELM_pls_ performed even worse than NCLELM. One possible reason is that some of the NCLELM base models for the ensemble construction of EnNCLELM_pls_ were not well learnt, due to the insufficiency of training samples in the large-value zone, and, thus, the combination of such models resulted in a performance deterioration instead of improvement. By contrast, the proposed EnSSNCLELM approach still obtained the most competitive prediction accuracy in the large-value zone, which was mainly because the inclusion of high-confidence pseudo-labeled data significantly improved the generalization capability of the SSNCLELM base models.

In addition, Figure 11 presents the performance of the individual models before and after incorporating semi-supervised learning for the industrial fed-batch CTC fermentation process. One can see that SSNCLELM base models achieved a better performance than the NCLELM base models, except for a few cases, which is attributed to the augmentation of the labeled training data, resulting from the good pseudo-label data. After introducing the stacking combination, EnNCLELM_pls_ delivered a performance close to that of the best base model, while EnSSNCLELM produced much better predictions than all the base models. In comparison, both EnNCLELM_avg_ and EnSSNCLELM_avg_ did not deliver a prediction accuracy enhancement. This was mainly because the simple averaging rule cannot consider the differences among the base models. Overall, the competitive performance of EnSSNCLELM was mainly due to the introduction of semi-supervised learning for enhancing the accuracy and diversity of the base models, as well as the utilization of ensemble learning, to further improve the prediction accuracy and reliability.

The above case study results further indicate the effectiveness of the proposed MLPLO technique in extending the labeled training data and verify the superiority of the proposed EnSSNCLELM method over traditional soft sensor methods, for providing accurate quality variable predictions in industrial processes, especially for application scenarios with limited labeled data.

## 5. Conclusions

A novel semi-supervised soft sensor modeling method, referred to as EnSSNCLELM, was proposed for the quality variable prediction of industrial processes, where the labeled data are scarce but the unlabeled data are rich. The proposed approach aims to enhance the inferential performance by leveraging both the labeled and unlabeled data, as well as combining semi-supervised and ensemble learning paradigms to achieve complementary strengths. First, to reduce the prediction instability and improve the prediction accuracy of ELM, the NCLELM algorithm was developed by introducing the NCL rule into ELM modeling. In this way, the diversity among the base ELM models is explicitly considered during NCLELM modeling. Second, by using NCLELM as the base learning technique, we proposed a MLPLO approach to provide high-confidence pseudo-labeled data, where the issue of estimating pseudo labels is formulated as an explicit optimization problem and solved through evolutional optimization. In addition, based on the extended labeled data, a set of diverse SSNCLELM models were built. Finally, these models were combined using a stacking strategy, after ensemble pruning according to the performance enhancement of semi-supervised models against supervised models. The application results from two chemical processes indicated that the proposed algorithm allows for a significant performance improvement over the traditional supervised and semi-supervised soft sensor methods.

The success of EnSSNCLELM is highly dependent on obtaining high-quality pseudo-labeled data, as well as building and combing diverse and accurate semi-supervised base models. Thus, how to formulate and solve the PLOP problem deserves further studies, e.g., defining more effective optimization objectives and employing multi-objective pseudo-labeling optimization, instead of a single-objective method. Furthermore, it is of great importance to ensure safe learning of the base models, i.e., avoiding reducing learning performance significantly when using unlabeled data. This may be achieved through efficiently selecting well-performing base models for ensemble construction and utilizing informative unlabeled data for semi-supervised modeling. These issues are left to be investigated in the near future.

## Figures and Tables

**Figure 1 sensors-21-08471-f001:**
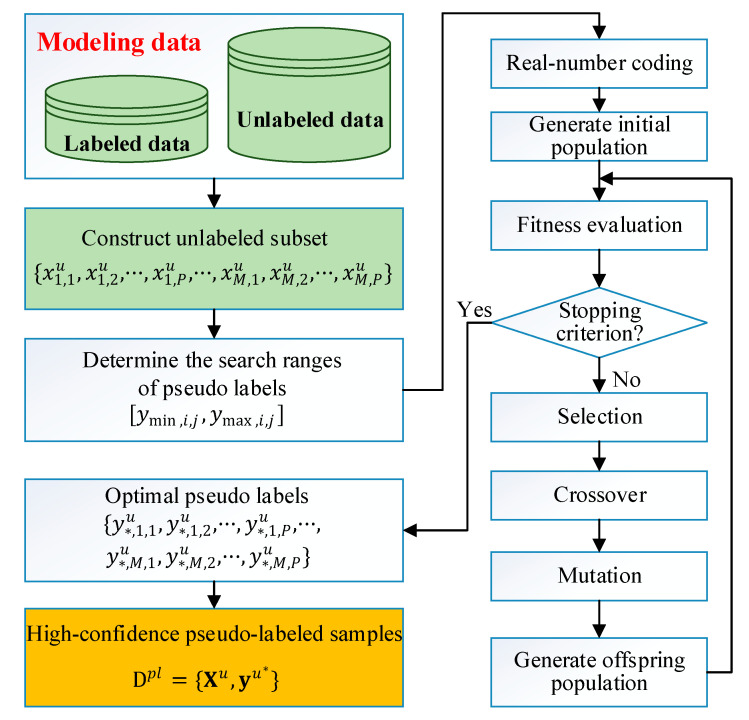
Schematic diagram of solving the MLPLO problem using GA.

**Figure 2 sensors-21-08471-f002:**
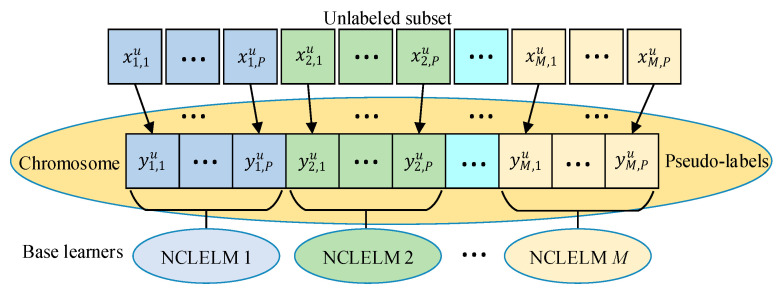
Individual structure for GA-based pseudo-labeling optimization.

**Figure 3 sensors-21-08471-f003:**
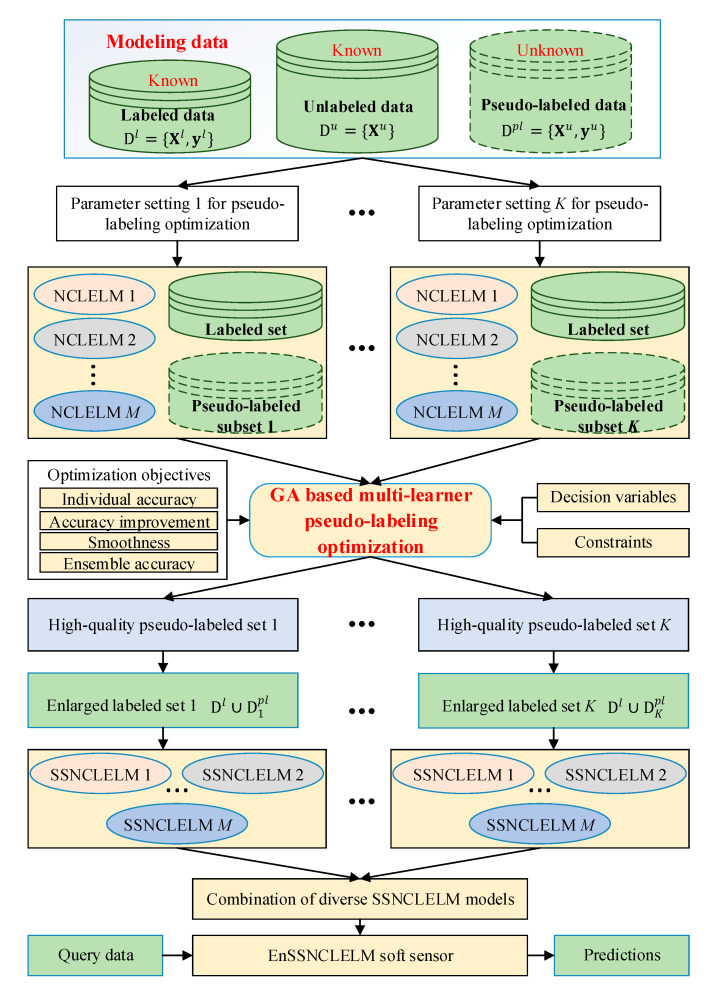
Workflow of the proposed EnSSNCLELM soft sensor.

**Figure 4 sensors-21-08471-f004:**
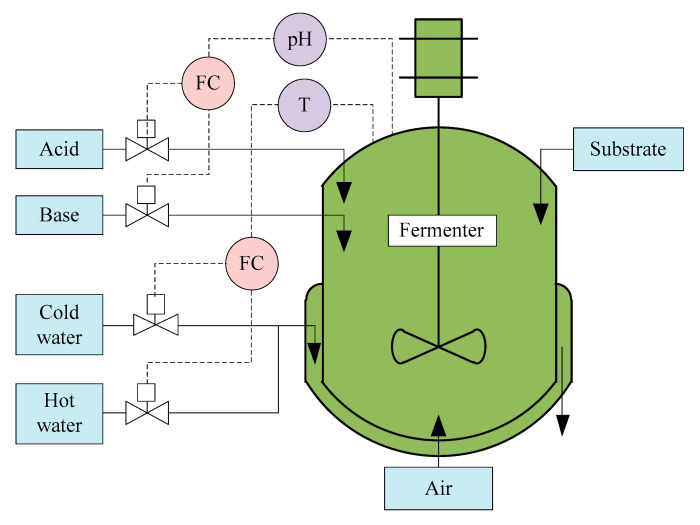
Flow diagram of penicillin fermentation process.

**Figure 5 sensors-21-08471-f005:**
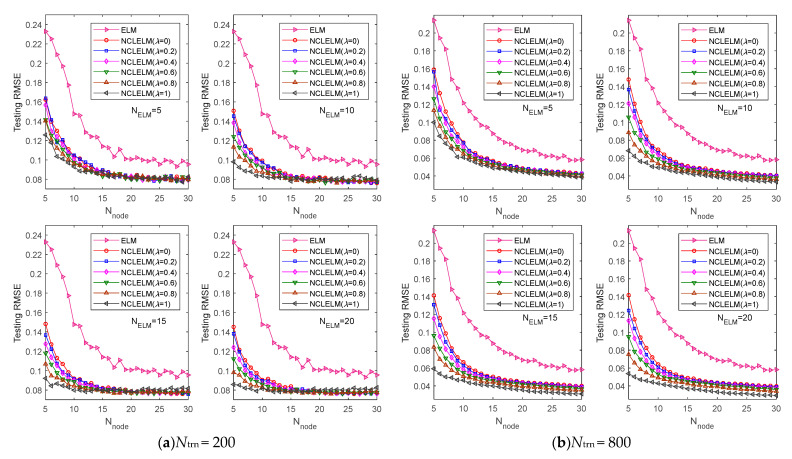
Prediction results of the ELM, EnELM, and NCLELM methods under different parameter settings. In (**a**,**b**), the training sample size was *N*_trn_ = 200 and 800, respectively.

**Figure 6 sensors-21-08471-f006:**
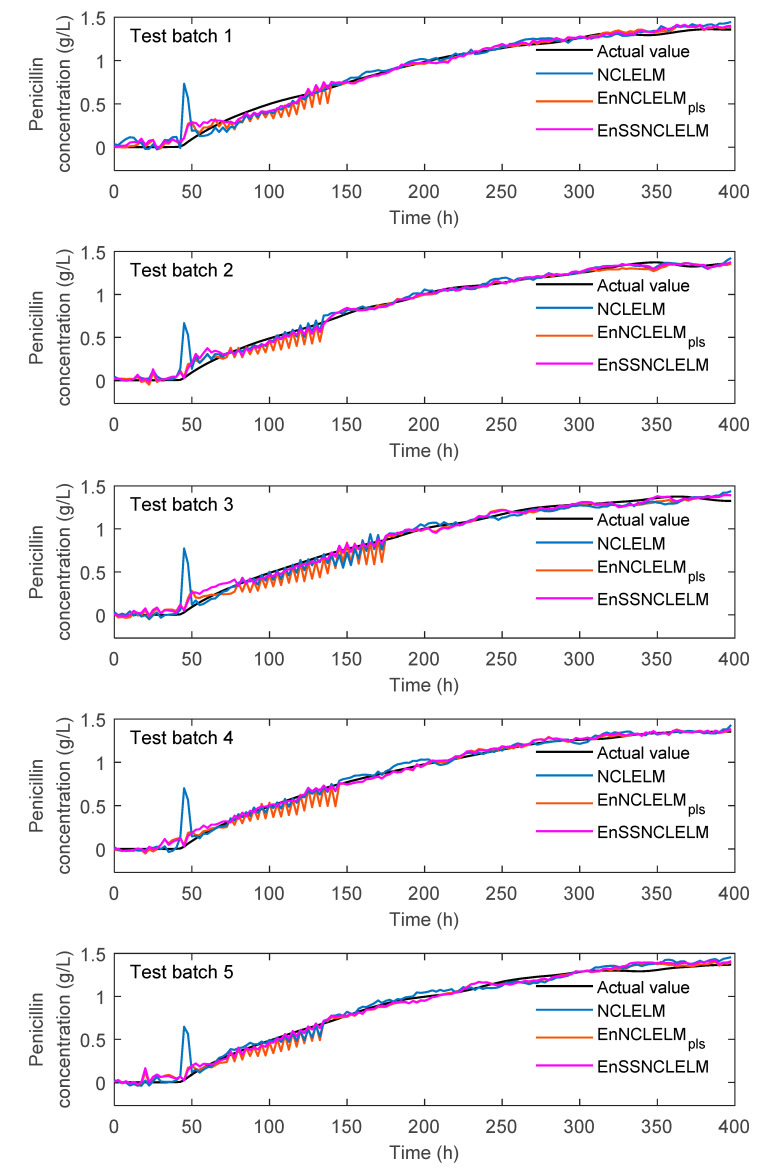
Trend plots of penicillin concentration predictions using ELM, EnNCLELM_pls_, and EnSSNCLELM methods in penicillin fermentation process.

**Figure 7 sensors-21-08471-f007:**
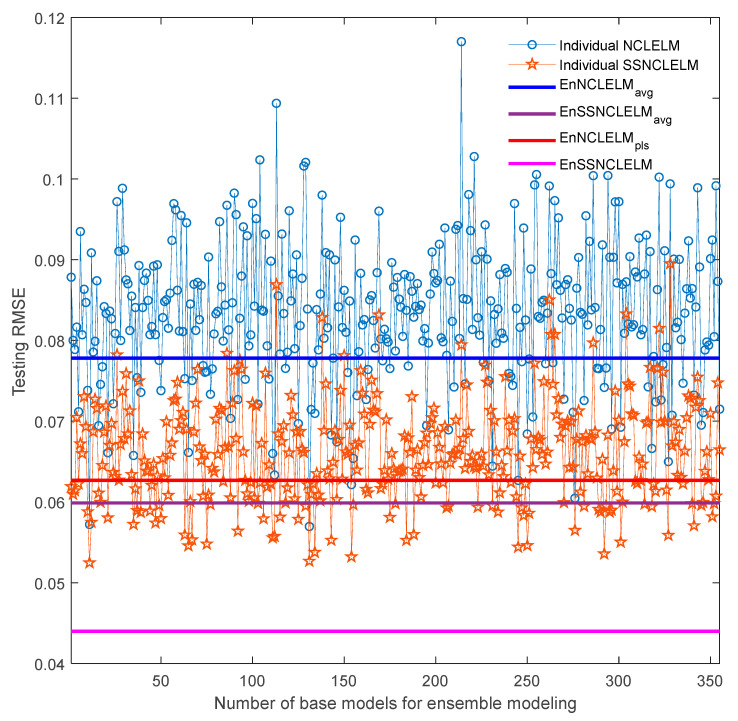
Prediction performance of individual models before and after incorporating semi-supervised learning.

**Figure 8 sensors-21-08471-f008:**
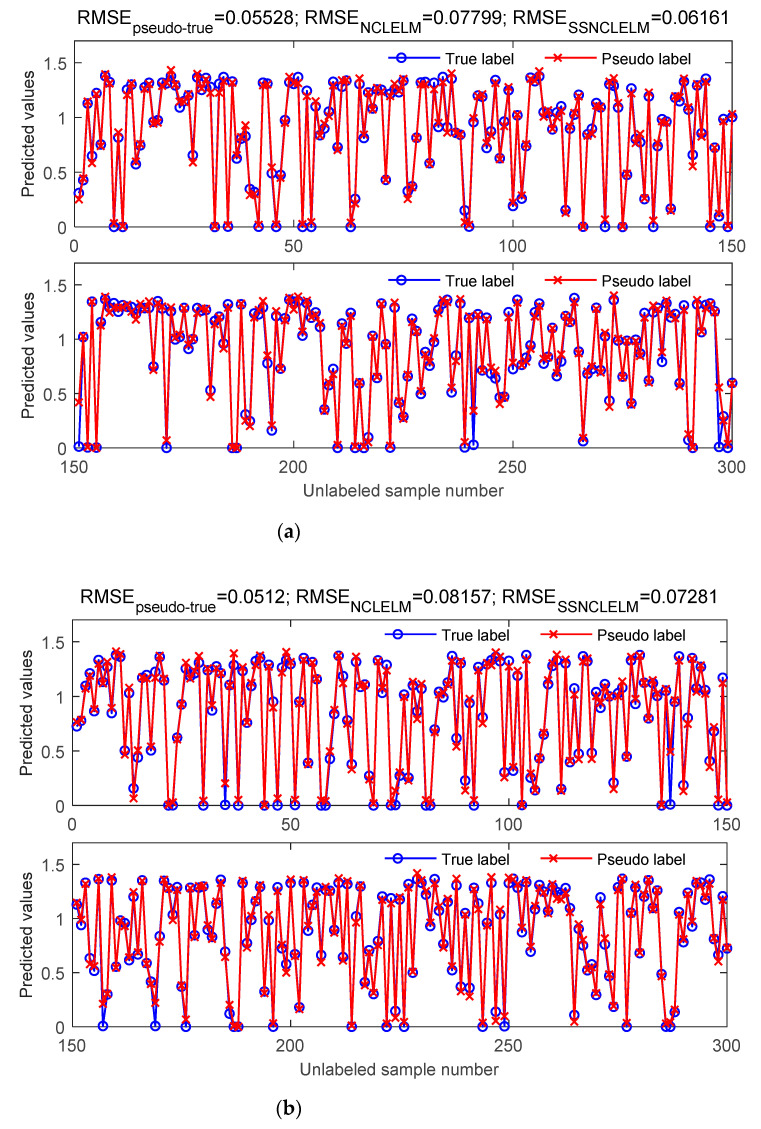
Comparison of the real and pseudo labels estimated by the proposed pseudo-labeling optimization method from (**a**) Example run 1, and (**b**) Example run 2. RMSE_pseudo–true_ is the fitting accuracy between the pseudo and true labels, and RMSE_NCLELM_ and RMSE_SSNCLELM_ denote the prediction accuracy of the NCLELM, before and after including the pseudo-labeled data, respectively.

**Figure 9 sensors-21-08471-f009:**
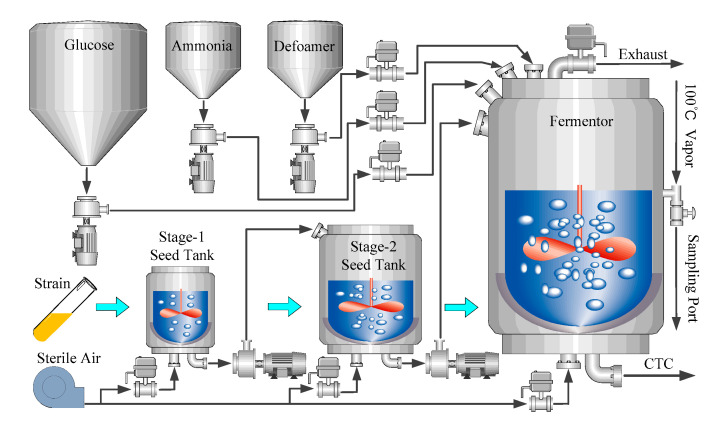
Flow diagram of an industrial fed-batch CTC fermentation process.

**Figure 10 sensors-21-08471-f010:**
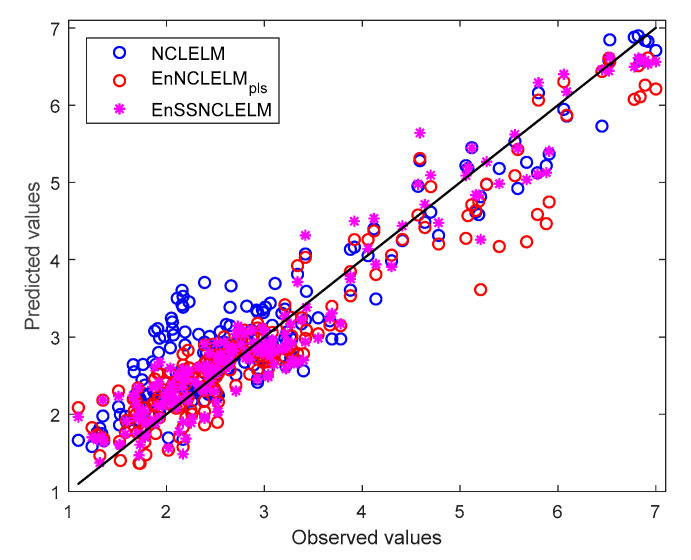
Scatter plots of the observed and predicted substrate concentration using the NCLELM, EnNCLELM_pls_, and EnSSNCLELM methods.

**Figure 11 sensors-21-08471-f011:**
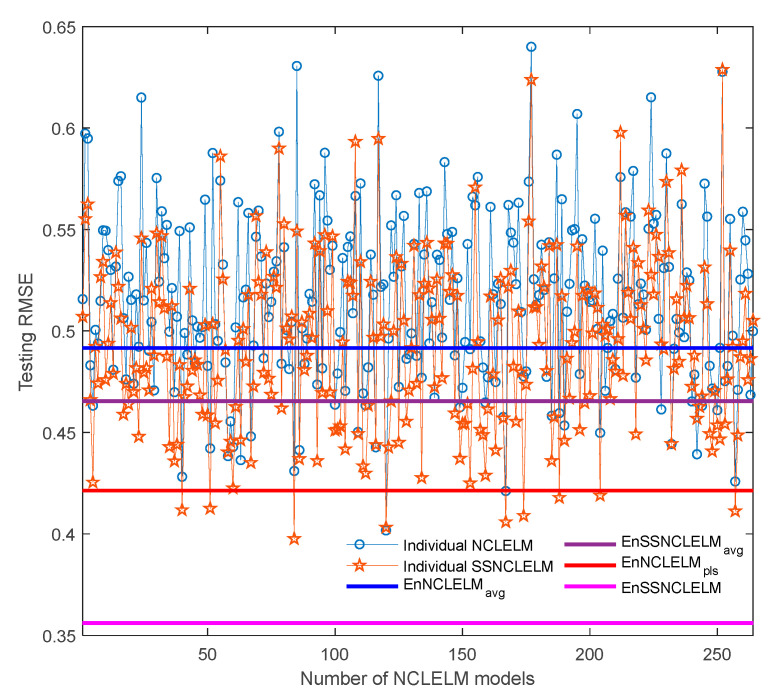
The performance of individual models before and after incorporating semi-supervised learning for industrial fed-batch CTC fermentation process.

**Table 1 sensors-21-08471-t001:** Input variables for soft sensor development in the penicillin fermentation process.

No.	Variable Description (Unit)
1	Culture time (h)
2	Aeration rate (L/h)
3	Agitator power (W)
4	Substrate feed rate (L/h)
5	Substrate feed temperature (K)
6	Dissolved oxygen concentration (g/L)
7	Culture volume (L)
8	Carbon dioxide concentration (g/L)
9	pH (−)
10	Fermenter temperature (K)
11	Generated heat (kcal)
12	Cooling water flow rate (L/h)

**Table 2 sensors-21-08471-t002:** Prediction results of penicillin concentration using different soft sensor methods in the penicillin fermentation process.

Method	RMSE	*R* ^2^
ELM	0.1044	0.9479
NCLELM	0.0812	0.9689
CoELM	0.0772	0.9720
SSELM	0.1019	0.9506
EnNCLELM_avg_	0.0778	0.9714
EnSSNCLELM_avg_	0.0599	0.9822
EnNCLELM_pls_	0.0627	0.9821
EnSSNCLELM	0.0440	0.9908

**Table 3 sensors-21-08471-t003:** Input variables for soft sensor development in the industrial CTC fermentation process.

No.	Variable Description	No.	Variable Description
1	Cultivation time (min)	6	Volume of air consumption (m^3^)
2	Temperature (°C)	7	Substrate feed rate (L/h)
3	pH	8	Volume of substrate consumption (L)
4	Dissolved oxygen concentration (%)	9	Volume of ammonia consumption (L)
5	Air flow rate (m^3^/h)	10	Volume of culture medium (m^3^)

**Table 4 sensors-21-08471-t004:** Prediction results of the various soft sensor methods for the CTC fermentation process.

Method	RMSE	*R* ^2^
ELM	0.5993	0.8009
NCLELM	0.5214	0.8527
CoELM	0.5033	0.8614
SSELM	0.5566	0.8303
EnNCLELM_avg_	0.4916	0.8699
EnSSNCLELM_avg_	0.4654	0.8834
EnNCLELM_pls_	0.4214	0.9044
EnSSNCLELM	0.3561	0.9317

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
