# Peer review of "Pseudo-Labeling Optimization Based Ensemble Semi-Supervised Soft Sensor in the Process Industry"

_sensors, 2021, doi:10.3390/s21248471_

Round 1

Reviewer 1 Report

The paper proposes a novel semi-supervised machine learning method, called EnSSNCLELM, for soft sensors and demonstrates its industrial applications. I’ve found the benefits below and recommend to the editorial board to accept with a minor revision.

+ well described background information regarding the state-of-the-art

+ well described the process of the proposed method

+ clear representation of the results with an extensive comparison study

Minor formatting issues need to be corrected

Texts in blue color without their hyperlink should be black (lines 552, 808, 814, 826, 839, 847, 849, 850, 863, 877, 879, 880, 901, 903, and 917)

Reference citation should be more organized in order of appearing in the document.

e.g. [16, 17, 20] without previous [18], [19]. And missing citation of [19].

Reviewer 2 Report

It is a common problem in process industry that it is hard to obtain online measurements of certain quantities that are relevant for process control. Traditional approaches to estimate diffcult-to-measure variables combine 1st principle models with available sensor data (e.g. observers, Kalman filters). The problem is to obtain reliable process models. In this article,  the authors follow an alternative purely data driven approach, using machine learning techniques. Starting from a more common method called "Extreme Learning Machine" (ELM), they introduce various improvements and end up with an algorithm they call "ensemble semi-supervised negative correlation learning extreme learning machine"
(EnSSNCLELM) (I wonder if there might be a more intuitive name for this). The mathematical derivation is presented in detail and seems to be sound to me.
The authors use two examples from pharmaceutical industry to test the proposed methods. In both examples, they reconstruct hard-to-measure concentrations (of penicillin and substrate) from easily accessible process data. They show that their methods work well and are an improvement compared to existing approaches.
In my opinion, this is interesting and innovative work worth publishing. I only have a few minor suggestions for improvement:
- The definition of the function g(.) in Eq. (1) and the paragraph below ist not quite consistent, as g(.) sometimes has a scalar and sometimes a vectorial argument.
- typo in Fig. 1: "serch" instead of "search"
